# RiskPO: Risk-Based Policy Optimization via Verifiable Reward for LLM Post-Training

**Tao Ren**[1]* **Jinyang Jiang**[1]* **Hui Yang**[1]* **Wan Tian**[1] **Minhao Zou**[1] **Guanghao Li**[2]
**Zishi Zhang**[1] **Qinghao Wang**[1] **Shentao Qin**[2,3] **Yanjun Zhao**[4] **Rui Tao**[1]
**Hui Shao**[5]† **Yijie Peng**[1,6]†
[1] Peking University    [2] Tsinghua University    [3] OriginFlow    [4] Xi'an Jiaotong University
[5] Zhejiang University    [6] Xiangjiang Laboratory

## Abstract

Reinforcement learning with verifiable reward has recently emerged as a central paradigm for post-training large language models (LLMs); however, prevailing mean-based methods, such as Group Relative Policy Optimization (GRPO), suffer from entropy collapse and limited reasoning gains, which stem from overemphasizing high-probability output sequences while neglecting rare but informative reasoning paths. To address these challenges, we propose Risk-based Policy Optimization (RiskPO), which substitutes classical mean-based objectives with principled risk measures. Specifically, we introduce a Mixed Value-at-Risk objective that integrates weighted attention over multiple regions of the reward distribution, thereby amplifying gradient signals on challenging instances and preventing overconfident convergence. We further design a bundling scheme that aggregates multiple questions into bundles, thus enriching the feedback signal and yielding more stable and informative training dynamics. Theoretically, we prove that the risk-averse update alleviates entropy collapse and promotes exploration. Numerically, RiskPO achieves consistent and significant improvements in mathematical reasoning, multi-modal reasoning, and code generation benchmarks, surpassing GRPO and its variants on both Pass@1 and Pass@k metrics. Our results demonstrate that risk-based optimization provides a rigorous and effective paradigm for enhancing LLM reasoning capabilities. The implementation is available at https://github.com/RTkenny/RiskPO.

## 1 Introduction

Since reinforcement learning (RL) provides a unified framework that flexibly accommodates diverse training targets and feedback, it has become a key technique for the post-training of large language models (LLMs). Based on such a foundation, RL with verifiable reward (RLVR) has recently been recognized as an effective paradigm for enhancing the reasoning ability of LLMs. Unlike traditional RL from human feedback, it leverages objective and binary reward signals, providing clear optimization feedback. Maximizing the expected average reward is anticipated to improve task performance of LLMs. Within this framework, a series of efficiency-oriented extensions have been developed from the classical policy-based RL method. Among them, Group Relative Policy Optimization (Shao et al., 2024; Guo et al., 2025, GRPO) achieves substantial efficiency gains by discarding redundant structures originally designed for standard RL tasks, and has

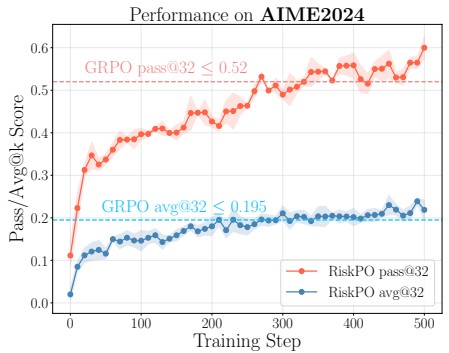

Figure 1: Pass@32 and Avg@32 learning curves of DeepSeek-R1-Distill-Qwen-1.5B trained by RiskPO on AIME2024.

---

* Equal contribution.
† Corresponding author: huishao@intl.zju.edu.cn, pengyijie@pku.edu.cn.

become the de facto baseline in this area. Since then, several GRPO variants have been proposed; see Section 2 for details.

However, RLVR methods that maximize average performance suffer from the fundamental issue of entropy collapse. Prior work shows that models trained via RLVR often experience rapid entropy collapse in the early stages of training, leading to premature convergence and a plateau in performance with little subsequent improvement (Cui et al., 2025; Gao et al., 2025). Entropy, as emphasized by several studies, serves as a key indicator of exploration capacity in RL (Wang et al., 2025b; Cheng et al., 2025; Hou et al., 2025). Once entropy collapses, the model becomes overconfident, reduces exploration prematurely, and fails to acquire new knowledge effectively. This constrained exploration ultimately limits its reasoning capabilities and overall performance. As a consequence, LLMs do not truly expand their intrinsic reasoning capacity or boundary; the observed improvements often reflect a more efficient sampling of known answers rather than genuinely stronger reasoning skills (Yue et al., 2025a; Xiong et al., 2025; Chen et al., 2025; Gao et al., 2025). This boundary effect implies that GRPO may only enhance short-horizon performance metrics (e.g., Pass@1) without significantly lifting the capability of the base model.

We argue that a key reason behind these challenges is that GRPO employs the mean as its objective, which is inherently misaligned with the goal of improving reasoning ability. A mean-based objective disproportionately emphasizes common, high-probability generation paths while neglecting rare yet informative reasoning trajectories, leading to premature convergence and limited exploration. Even worse, if the model consistently generates all wrong answers to a question, the estimated GRPO advantage collapses to zero, leaving the model without any effective learning signal in its weakest areas. Over-concentrating gradient updates on easier questions brings only marginal gains, as it mainly reinforces what the model already knows instead of improving its ability on harder problems. In contrast, risk-averse optimization objectives, such as Conditional Value-at-Risk (CVaR) or Range Value-at-Risk (RVaR), can encourage the model to explore difficult problems and enhance reasoning abilities. By amplifying gradient signals from low-reward answers, these objectives natu-

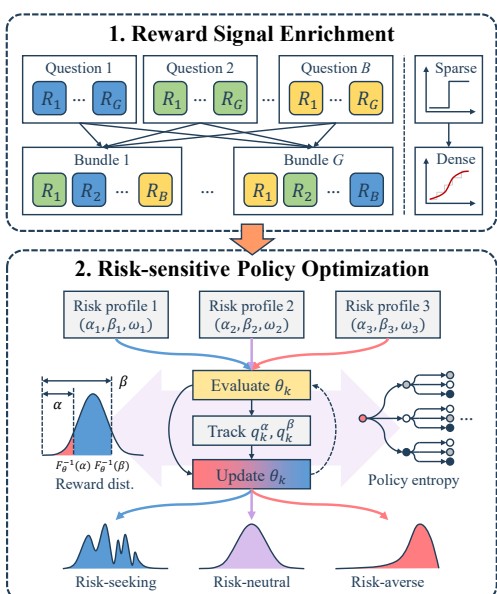

Figure 2: The framework of RiskPO.

rally encourage the policy to reduce overconfidence, diversify its search, and promote novel reasoning strategies. Consequently, risk-based objectives provide an effective handle for better mitigating entropy collapse, preventing overfitting to easy problems, and driving genuine improvements of the reasoning boundary.

We propose Risk-based Policy Optimization (RiskPO), which employs a novel risk-sensitive objective termed Mixed Value-at-Risk (MVaR). Compared with mean-based post-training methods, our risk-based approach demonstrates superior performance in encouraging exploration and fostering stronger reasoning capabilities. The overall framework of RiskPO is illustrated in Figure 2. We summarize our contributions as follows:

1. To the best of our knowledge, we are the first to incorporate risk measures into the training objective. Since the reward for a single question is binary, we propose grouping multiple questions into a bundle to enrich the feedback signal. It is shown to avoid the zero advantage issue and strengthen gradient signals for hard problems, thereby facilitating exploration.

2. We provide theoretical results that explain the superiority of the proposed MVaR objective. By analyzing the entropy mechanism, we demonstrate that the risk-averse configuration can effectively mitigate entropy collapse.

3. We conduct extensive numerical experiments to evaluate the performance of our algorithm. RiskPO consistently outperforms GRPO and other baselines on multiple mathematical reason-

ing tasks. On Pass@k metrics, RiskPO even achieves better performance, indicating its strong capacity for exploration and acquisition of new reasoning skills.

## 2 RELATED WORKS

**RL for LLM Post-Training.** RL has played a critical role in the post-training phase of LLM (Shao et al., 2024; Christiano et al., 2017; Lambert et al., 2022). Through verifiable reward, the LLM learn complicated reasoning skills across solving math problems and coding (Zhao et al., 2025a; Chen et al., 2024; Huang et al., 2025). Originating from the PPO, much literature has proposed new methods to cater to the requirements of RLVR. ReMax (Li et al., 2024) proposes to use the deterministic output of the LLM as the baseline to reduce variance. DAPO (Yu et al., 2025) incorporates four engineering tricks to improve GRPO. VAPO shows that the value-based RL method can also perform well in RLVR (Yue et al., 2025b). GPG (Chu et al., 2025) investigates the normalizing factor in GRPO. Several literatures (Wang et al., 2025b; Cui et al., 2025; Cheng et al., 2025; Wang et al., 2025a) investigate the entropy mechanism in RLVR, pointing out the significance of exploration in RLVR. GSPO (Zheng et al., 2025) and GMPO (Zhao et al., 2025b) focus on stabilizing the RL training. ProRL (Liu et al., 2025a) shows that with stabilized training, the performance gain would have a log-scale relationship with the training time.

**Risk-Sensitive RL.** Risk-sensitive RL (see, e.g., Ren et al., 2024; Petersen et al., 2019) seeks to shape the entire reward distribution rather than merely optimizing its mean. Chow et al. (2015) investigates the Markov Decision Process (MDP) under CVaR objective and proposes a dynamic-programming based solution. La & Ghavamzadeh (2013) and Prashanth et al. (2016) use finite difference to optimize risk measure under the MDP setting. Dabney et al. (2018b;a) introduces state-action value distribution approximation techniques to improve the effectiveness, which is referred to as distributional RL. CVaRPG (Tamar et al., 2015) and QPO (Jiang et al., 2022) design policy gradient style algorithm to optimize CVaR and quantile, respectively. Jiang et al. (2024) considers a more general case, optimizing the distortion risk measure in a policy gradient manner. There is also literature considering risk level as a constraint. Bertsekas (1997) use a Lagrangian approach to solve RL problems. Borkar & Jain (2014) use CVaR as a constraint, and Chow et al. (2018) develop actor-critic algorithms under quantile and CVaR constraints.

## 3 RETHINKING RLVR FROM A DISTRIBUTIONAL PERSPECTIVE

We formalize the post-training problem of RLVR as follows. Given an input problem $x$ sampled from a dataset $\mathcal{D}$, an LLM parameterized as $\pi_\theta$ generates a response $y \sim \pi_\theta(\cdot|x)$. A rule-based verifier $R(\cdot)$ then evaluates the correctness of the response, returning one if $y$ is correct and zero otherwise. Notably, no intermediate process-level feedback is provided. The standard objective in this setting is to maximize the expected reward: $\mathcal{J}(\theta) = \mathbb{E}_{x\sim\mathcal{D},\, y\sim\pi_\theta(\cdot|x)}[R(y)]$. With a score-function method, its gradient is given by $\nabla_\theta \mathcal{J}(\theta) = \mathbb{E}[R(y)\nabla_\theta \ln \pi_\theta(y|x)]$, resulting in a standard RL framework, where a baseline or so-called value model is used for variance reduction.

As a widely adopted baseline for RLVR, GRPO (Shao et al., 2024) replaces the value model with sequence-level standardized rewards computed within a group of responses. We denote by $y_{<t}$ the partial response consisting of the first $t$ tokens, i.e., $\pi_\theta(y|x) = \prod_t \tilde{\pi}_\theta(y_t|x, y_{<t})$. Specifically, given a query $x$ and a group of $G$ responses $\{y_i\}_{i=1}^G$ sampled from a reference model $\pi_{\theta'}(\cdot|x)$, the GRPO objective is defined as

$$\mathcal{J}_{\text{GRPO}}(\theta) = \mathbb{E}_{\substack{x\sim\mathcal{D} \\ \{y_i\}_{i=1}^G \sim \pi_{\theta'}(\cdot|x)}} \left[ \frac{1}{G} \sum_{i=1}^G \frac{1}{|y_i|} \sum_{t=1}^{|y_i|} \min\left( w_{i,t}(\theta)\,\hat{A}_i,\ \text{clip}(w_{i,t}(\theta), 1-\epsilon, 1+\epsilon)\,\hat{A}_i \right) \right],$$

where $\hat{A}_i = \frac{R(y_i) - \frac{1}{G}\sum_{j=1}^G R(y_j)}{\text{Std}(\{R(y_j)\}_{j=1}^G)}$ denotes the standardized feedback, while $w_{i,t}(\theta) = \frac{\tilde{\pi}_\theta(y_{i,t}|x, y_{i,<t})}{\tilde{\pi}_{\theta'}(y_{i,t}|x, y_{i,<t})}$ is the importance sampling ratio that enables multiple parameter updates per group of generated data. Despite these modifications, GRPO remains fundamentally a method for optimizing the mean performance of LLMs. Since the reward provided by the verifier is an indirect objective, we argue it may not be the best practice for RLVR to optimize its expectation. Instead, we propose to adopt a distributional perspective. The most challenging problems correspond to the left tail of the reward

distribution. These samples represent the questions that the model has not yet mastered. Such hard cases often lead to gradient vanishing in GRPO. For example, when all responses are incorrect, the computed advantage collapses to zero, which provides no meaningful training signal. As a result, the model fails to improve on its weakest regions of the distribution.

Therefore, beyond optimizing the expectation, we claim that it is more beneficial to consider the distributional structure of performance, particularly the lower tail. Incorporating risk measures, such as CVaR or RVaR, into the training objective emphasizes hard problems in the tail of the distribution and provides a finer-grained and more robust learning signal for RLVR.

# 4 MASTERING THE UNCERTAINTY VIA RISK-BASED POLICY OPTIMIZATION

In this section, we introduce our risk-based objective for RLVR and the associated post-training methodology, establishing a principled framework for enhancing LLM reasoning ability.

Denote the RLVR reward signal distribution by $F_\theta(\cdot)$, where the parameter $\theta$ reflects the stochasticity induced by the LLM $\pi_\theta(\cdot|x)$. RVaR is defined to capture the average performance within a specified quantile interval of the distribution. Let $F_\theta^{-1}(\alpha)$ be the $\alpha$-level quantile of $R(y)$. Then, for $0 \leq \alpha < \beta \leq 1$, RVaR on the interval $[\alpha, \beta]$ is written as

$$\mathcal{J}_{\text{RVaR}_{\alpha:\beta}}(\theta) := \mathbb{E}\big[R(y)|R(y) \in [F_\theta^{-1}(\alpha), F_\theta^{-1}(\beta)]\big] = \frac{1}{\beta - \alpha} \int_{F_\theta^{-1}(\alpha)}^{F_\theta^{-1}(\beta)} r dF_\theta(r), \quad (1)$$

that is, the conditional expectation of $R(y)$ given that it falls between its $\alpha$- and $\beta$-quantiles. To optimize the RVaR through gradient descent algorithms, we first derive the gradient of RVaR as shown in Theorem 1. The proofs of all theoretical results in this work can be found in Appendix B.

**Theorem 1.** *Assume $F_\theta(r)$ is continuously differentiable with respect to both the parameter $\theta$ and the variable $r$; the density is positive at the quantiles, i.e., $f_\theta(F_\theta^{-1}(\alpha)) > 0$ and $f_\theta(F_\theta^{-1}(\beta)) > 0$; and that the differentiation under the integral sign is justified. Then the gradient of RVaR is given by*

$$\nabla_\theta \mathcal{J}_{\text{RVaR}_{\alpha:\beta}}(\theta) = \frac{1}{\beta - \alpha} \mathbb{E}\big[g\big(R(y), F_\theta^{-1}(\alpha), F_\theta^{-1}(\beta)\big) \nabla_\theta \ln \pi_\theta(y|x)\big],$$

*where $g(z, a, b) = (z - a)^+ - (z - b)^+ + a - b$, and $(z)^+ = \max\{z, 0\}$.*

Note that when $\alpha = 0$, RVaR coincides with the lower-tail CVaR at level $\beta$, and the gradient in Theorem 1 reduces to $\nabla_\theta \mathcal{J}_{\text{RVaR}_{0:\beta}}(\theta) = \beta^{-1} \mathbb{E}\big[ - (F_\theta^{-1}(\beta) - R(y))^+ \nabla_\theta \ln \pi_\theta(y|x)\big]$. Since RVaR effectively places a window for control on the reward distribution, it is natural to further combine several such segments to better shape the overall distribution. Accordingly, we introduce a new objective into RLVR, namely Mixed Value-at-Risk (MVaR), which integrates metrics over multiple distributional segments as follows:

$$\mathcal{J}_{\text{MVaR}_{\alpha:\beta}^\omega}(\theta) = \left\{ (1 + \omega) \int_{F_\theta^{-1}(0)}^{F_\theta^{-1}(\alpha)} + \int_{F_\theta^{-1}(\alpha)}^{F_\theta^{-1}(\beta)} \right\} r dF_\theta(r), \quad (2)$$

where $\omega \geq 0$ controls the emphasis placed on tail samples during optimization, and high-performance samples are excluded from the current training process. Note that $\mathcal{J}_{\text{MVaR}_{\alpha:\beta}^\omega}(\theta) = (1 + \omega)\alpha \mathcal{J}_{\text{RVaR}_{0:\alpha}}(\theta) + (\beta - \alpha)\mathcal{J}_{\text{RVaR}_{\alpha:\beta}}(\theta)$. The gradient of (2) can be derived by Theorem 1.

However, the distributional information from a single question $x$ is limited, since the feedback is binary and offers only coarse signals. To obtain a richer source of information, we propose to group several questions into a bundle, i.e., $X := \{x_i\}_{i=1}^B \sim \mathcal{D}^{\otimes B}$, and calculate the advantage based on the sum of the individual question scores within the bundle. This aggregation transforms sparse binary feedback into a more informative distribution over bundle scores, enabling finer distinctions between different levels of performance and avoiding zero gradient on difficult questions. We then focus on optimizing the MVaR of the bundle score:

$$\mathbb{E}_{X \sim \mathcal{D}^{\otimes B}, \{y^i \sim \pi_\theta(\cdot|x_i)\}_{i=1}^B} \left[ R_B\big((1 + \omega)\mathbf{1}_{\{R_B \leq F_\theta^{-1}(\alpha)\}} + \mathbf{1}_{\{F_\theta^{-1}(\alpha) < R_B \leq F_\theta^{-1}(\beta)\}}\big) \right],$$

where $R_B = \sum_{i=1}^B R(y^i)$ denotes the bundle score. For each $i \in \{1, \ldots, B\}$, we sample $Y_i := \{y_j^i\}_{j=1}^G$ with $y_j^i \sim \pi_\theta(\cdot|x_i)$ i.i.d., and define $Y := \{Y_i\}_{i=1}^B$. Then we can generate $G$ bundles without overlaps from the $G \times B$ responses of $B$ questions. The gradient can be calculated by

$$\mathbb{E}_{X \sim \mathcal{D}^{\otimes B}, \{y_j^i\}_{j=1}^G \sim \pi_\theta(\cdot|x_i), \xi_i \sim \text{Unif}(\mathfrak{S}_G)} \left[ \frac{1}{G} \sum_{j=1}^G A_j \frac{1}{B} \sum_{i=1}^B \nabla_\theta \ln \pi_\theta(y_{\xi_{i,j}}^i | x_i) \right],$$

where $A_j = -(1+\omega)(F_\theta^{-1}(\alpha) - R_{B_j})^+ + g(R_{B_j}, F_\theta^{-1}(\alpha), F_\theta^{-1}(\beta))$ is the bundle-wise advantage under MVaR objective, $R_{B_j} = \sum_{i=1}^B R(y_{\xi_{i,j}}^i)$ is the bundle-wise score, $\xi$ is a permutation of $\{1, \ldots, G\}$ that independently draw $\xi_i \sim \text{Unif}(\mathfrak{S}_G)$ for every $i$, $\mathfrak{S}_G$ is the symmetric group on $G$ element, and $\xi_{i,j}$ is the $j$-th elements in the permutation. This construction yields $G$ disjoint bundles: the $j$-th bundle uses $\{y_{\xi_{i,j}}^i\}_{i=1}^B$, so that for each fixed $i$, $\{y_{\xi_{i,1}}^i, \ldots, y_{\xi_{i,G}}^i\}$ is a permutation of $\{y_1^i, \ldots, y_G^i\}$, i.e., every answer is used only once (without replacement).

To ensure stable improvement (Schulman et al., 2015; 2017) with multiple updates per bundle-wise MVaR objective evaluation, we adopt a trust-region style update with clipping and sequence-level importance sampling (Zheng et al., 2025). Since the reward in RLVR is only available at the sequence level, i.e., $y^i$, it is natural to define importance weights also at the sequence (response) level and then aggregate them into the bundle objective. Formally, given $B$ problems $X = \{x_i\}_{i=1}^B$ and $G$ responses per problem $Y_i = \{y_j^i\}_{j=1}^G$, we independently draw $\xi_i \sim \text{Unif}(\mathfrak{S}_G)$ for each $i$, yielding $G$ bundles: $\mathcal{P}_j = \{y_{\xi_{i,j}}^i\}_{i=1}^B, j = 1, \ldots, G$, where every responses is used without replication. We then construct the clipped MVaR objective at the bundle level, which constitutes the final loss for backpropagation:

$$\mathcal{J}_{\text{MVaR}}^{\text{clip}}(\theta) = \mathbb{E}_{X,Y,\{\xi_i\}} \left[ \frac{1}{G} \sum_{j=1}^G \frac{1}{B} \sum_{i=1}^B \min\left( s_j^i(\theta) A^{(j)}, \text{clip}(s_j^i(\theta), 1 - \epsilon, 1 + \epsilon) A^{(j)} \right) \right], \quad (3)$$

where $s_j^i(\theta) = \left( \frac{\pi_\theta(y_{\xi_{i,j}}^i | x_i)}{\pi_{\theta'}(y_{\xi_{i,j}}^i | x_i)} \right)^{1/|y_{\xi_{i,j}}^i|}$ is the sequence-wise importance sampling ratio.

Every token within the same bundle shares the same MVaR-based advantage $A^{(j)}$, ensuring that optimization is aligned with the unit of reward (the bundle score) and directs training toward the left tail of the performance distribution. We track $F_\theta^{-1}(\alpha)$ and $F_\theta^{-1}(\beta)$ in an online manner. After substituting the tracked quantiles into the advantage and deriving the gradient, we update model parameters accordingly. Therefore, RiskPO can be implemented as a two-timescale stochastic approximation algorithm. The pseudocode of the proposed algorithm is provided in Algorithm 1.

---

**Algorithm 1** Risk-Based Policy Optimization (RiskPO)

---

1: **Input**: quantile levels $\alpha, \beta$, weight $\omega$, policy $\pi_.$, learning rates $\{\gamma_k\}, \{\eta_k\}$, and iterations $K$
2: **Initialize**: policy parameter $\theta_0$, and quantile trackers $q_0^\alpha, q_0^\beta$
3: **for** $k = 1, \cdots, K$ **do**
4:     Sample $B$ questions, $X = \{x_i\}_{i=1}^B$, from the dataset $\mathcal{D}$
5:     Generate $G$ responses for each question, $\{y_j^i\}_{j=1}^G \sim \pi(\cdot|x_i)$ and evaluate the reward $R(y_j^i)$
6:     Sample from the symmetric group for $B$ times, $\xi_i \sim \text{Unif}(\mathfrak{S}_G)$, yielding $G$ bundles
7:     Track quantiles of bundle scores: $q_{k+1}^s = q_k^s + \gamma_k(s - \frac{1}{G} \sum_{j=1}^G \mathbf{1}\{R_{B_j} < q_k^s\}), s \in \{\alpha, \beta\}$
8:     Backpropagate the clipped MVaR objective (3) to compute $\nabla_\theta \mathcal{J}_{\text{MVaR}}^{\text{clip}}(\theta)$
9:     Update policy parameter: $\theta_{k+1} = \theta_k + \eta_k \nabla_\theta \mathcal{J}_{\text{MVaR}}^{\text{clip}}(\theta)$
10: **end for**
11: **Output**: Final policy parameter $\theta_{K+1}$

---

## 5 ENTROPY MECHANISM FOR RISK-SENSITIVE OBJECTIVE

A well-known issue in GRPO is entropy collapse, where the policy entropy rapidly decreases during training. This premature reduction in entropy limits exploration of alternative reasoning paths,

thereby constraining performance improvement and reducing the likelihood of discovering correct solutions. In this section, we conduct a per-step analysis for the change of policy entropy in the optimization and give a theoretical guarantee that our RVaR policy gradient can mitigate the entropy collapse issue.

Following the standard framework in policy-gradient literature (see, e.g., Agarwal et al., 2021; Shani et al., 2020; Abbasi-Yadkori et al., 2019), we conduct theoretical analysis under a tabular softmax formulation with deterministic sequence-level rewards. Specifically, we consider an input set $\mathcal{X}$ and an output set $\mathcal{Y}$. The actor is parameterized by a matrix $\theta \in \mathbb{R}^{|\mathcal{X}| \times |\mathcal{Y}|}$, where each entry $\theta_{x,y}$ is the logit for choosing $y$ given $x$. The policy is thus $\pi_\theta(y|x) = \frac{\exp(z_{x,y})}{\sum_{u \in \mathcal{Y}_x} \exp(z_{x,u})}$, and its conditional entropy is $\mathcal{H}(\pi_\theta|x) = -\sum_y \pi_\theta(y|x) \log \pi_\theta(y|x)$. Let $A_\theta(x, y)$ be the advantage value associated with the chosen algorithm. We begin with the following proposition, which links entropy dynamics to the covariance between the advantage and the log-probability of the output.

**Proposition 1.** *Fix a prompt $x$ and a finite set of complete sequences $\mathcal{Y}_x$. Consider a natural-gradient step, i.e., $\theta_{k+1} = \theta_k + \eta\Delta_k$, where $\Delta_{k,x,y} = A_{\theta_k}(x, y)$ otherwise zero, then*

$$\mathcal{H}(\pi_{\theta_{k+1}}|x) - \mathcal{H}(\pi_{\theta_k}|x) = -\eta \operatorname{Cov}_{y \sim \pi_{\theta_k}(\cdot|x)}(\log \pi_{\theta_k}(y|x), A_{\theta_k}(x, y)) + O(\|\Delta_k\|^2). \quad (4)$$

The proposition indicates that the correlation between the advantage and the log probability of the output affects entropy changes: a positive correlation leads to entropy decrease, and vice versa. High advantage and high log probability suggest that the model is very confident about samples with high advantage. Therefore, over-optimizing already well-learned problems accelerates entropy collapse, reducing the model's opportunities for trial and error on difficult problems and ultimately limiting policy performance. RiskPO, which uses MVaR as its objective, can alleviate this issue. It focuses more on difficult problems, i.e., samples from the left tail of the reward distribution, and clips the gradient signal for well-learned problems. In the following theorems, we provide theoretical justification by analyzing the correlation between advantage and log-likelihood, showing that RiskPO induces higher-entropy policy updates. Before that, we present the following assumption, which captures the most intuitive scenario but is not the only suitable one. Let $\psi(r) = \mathbb{E}[\log \pi_\theta(y|x)|R = r]$ denote the conditional log-likelihood.

**Assumption 1.** *The conditional log-probability of output $\psi(r)$ is non-decreasing for $r \geq F_\theta^{-1}(\beta)$, non-increasing for $r \leq F_\theta^{-1}(\alpha)$, and $\psi(F_\theta^{-1}(\alpha)), \psi(F_\theta^{-1}(\beta)) \geq \mathbb{E}[\log \pi_\theta(y|x)]$.*

Intuitively, this assumption captures the behavior of a pre-trained base model. For relatively easier problems in the upper tail of the reward distribution, the model exhibits well-calibrated confidence, assigning higher probabilities to correct answers (upper-tail monotonicity). In contrast, for harder problems in the lower tail, the model often fails systematically: it allocates substantial probability mass to incorrect outputs, effectively making confident but consistently wrong predictions (lower-tail monotonicity). We empirically validate this assumption using DeepSeek-R1-Distill-Qwen-1.5B on the training set. For each question, the model generates 16 responses and computes the mean reward across these responses. Recall that the length-normalized sequence-level log-probability is de-

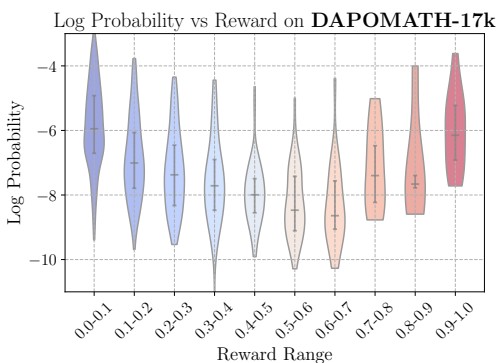

Figure 3: Log-probabilities as a function of reward quantile levels for DeepSeek-R1-Distill-Qwen-1.5B on DAPOMATH-17K.

fined as $\log \pi_\theta(y|x) = |y|^{-1} \sum_{t=1}^{|y|} \log \tilde{\pi}_\theta(y_t|y_{<t}, x)$. As shown in Figure 3, the sequence-wise logprob exhibits monotonicity in $[0, 0.3]$ and $[0.7, 1]$, which justifies Assumption 1. Denote the advantages in the associated algorithms as $A_{\text{MVaR}}$ and $A_{\text{Mean}}$, see Appendix B.3 for details. Next, we present a comparison of the correlations between the advantage values and the log-likelihood.

**Theorem 2.** *If Assumption 1 holds and $E[|\log \pi(y|x)|] < \infty$, then the covariance between MVaR-based advantages and output log-probabilities is smaller than that of mean-based methods, i.e.,*

$$\operatorname{Cov}_{y \sim \pi_\theta(\cdot|x)}(\log \pi_\theta(y|x), A_{\text{MVaR}_{\alpha:\beta}^\omega}) \leq \operatorname{Cov}_{y \sim \pi_\theta(\cdot|x)}(\log \pi_\theta(y|x), A_{\text{Mean}}). \quad (5)$$

The combination of Proposition 1 and Theorem 2 implies that, under the same update setting, RiskPO leads to higher output entropy compared with mean-based methods. Similarly, we can conclude that risk-seeking objectives assign greater weight to high-reward outcomes, thereby amplifying the covariance between log-probabilities and advantages. This stronger coupling causes entropy to decrease more rapidly and results in severe entropy collapse. To further elucidate the relationship between correlation and objective design, beyond the specified MVaR objective, we can generalize to a more general transformation of the reward or so-called advantage value. This general form allows us to assess the strength of the correlation and, in turn, reason about the resulting policy entropy. Please refer to Appendix B.4 for details.

## 6 EXPERIMENTS

To comprehensively evaluate the effectiveness of RiskPO, this section conducts systematic experiments across a broad spectrum of tasks, including mathematical reasoning, code generation, and multi-modal reasoning. We benchmark on more than ten datasets that span different levels of difficulty. Through these experiments, our goals are threefold: (i) to verify that risk-based objectives consistently outperform mean-based methods across domains, (ii) to demonstrate that the proposed distributional perspective leads to tangible improvements on the hardest reasoning problems, and (iii) to provide empirical evidence that RiskPO mitigates entropy collapse and enables genuine expansion of reasoning capability rather than merely improving sampling efficiency. Detailed experimental settings and supplementary results are provided in Appendix A.

### 6.1 MAIN RESULTS

Table 1 reports Pass@1 accuracy across six hard-level mathematical reasoning benchmarks. We observe that RiskPO consistently achieves the best performance among all methods, outperforming both the base models and recent GRPO variants. In particular, RiskPO attains an average score of 46.65, representing a +2.78 absolute improvement over the strongest baseline DAPO (43.87) and a +6.24 improvement over vanilla GRPO (40.41). The gains are especially pronounced on the most challenging AIME datasets, where RiskPO surpasses DAPO by nearly +6.7 points (33.3 vs. 26.6). These results demonstrate that emphasizing distributional risk through our MVaR objective substantially improves reasoning ability, not only enhancing performance on easier datasets like AMC and MATH500 but also pushing the frontier on the hardest Olympiad-style tasks.

Table 1: Pass@1 performance on hard-level mathematical reasoning benchmarks.

| Model | AIME25 | AIME24 | AMC | MATH500 | Minerva | Oly. | Avg. |
|---|---|---|---|---|---|---|---|
| Qwen2.5-Math-1.5B | 6.6 | 10.0 | 43.4 | 61.8 | 15.1 | 28.4 | 27.55 |
| Qwen2.5-Math-1.5B-Instruct | 10.0 | 10.0 | 48.2 | 64.2 | 26.5 | 35.2 | 32.35 |
| DeepSeek-R1-Distill-Qwen-1.5B | 13.3 | 13.3 | 32.5 | 59.8 | 20.3 | 30.5 | 28.28 |
| Dr.GRPO-1.5B (Liu et al., 2025b) | 20.0 | 23.3 | 54.7 | 77.4 | 26.3 | 38.1 | 39.97 |
| GRPO-1.5B (Shao et al., 2024) | 20.0 | 20.0 | 56.6 | 79.2 | 27.1 | 39.6 | 40.41 |
| GPG-1.5B (Chu et al., 2025) | 16.6 | 20.0 | 55.7 | 74.5 | 28.8 | 37.6 | 38.90 |
| DAPO-1.5B (Yu et al., 2025) | 30.0 | 26.6 | 58.6 | 78.2 | 29.2 | 40.6 | 43.87 |
| GMPO-1.5B (Zhao et al., 2025b) | 23.3 | 23.3 | 54.2 | 76.2 | 29.2 | 39.2 | 40.90 |
| RiskPO-1.5B (Ours) | **33.3** | **33.3** | **60.8** | **81.8** | **29.5** | **41.2** | **46.65** |

We complement these findings in hard-level math tasks with results on easier mathematical reasoning, multi-modal reasoning, and code generation tasks in Table 2. While performance gaps on GSM8K are naturally small due to the dataset's simplicity, RiskPO maintains a measurable advantage on MATH and yields consistent improvements on both LiveCodeBench and Geometry3K. Together, these results underscore the broad applicability of risk-sensitive objectives and their ability to enhance reasoning capacity across diverse domains.

We argue that the RiskPO is expanding the reasoning boundary of the base model. To support the claim, we present the evolution of Pass@1, Pass@8, and Pass@16 on AMC and MATH500 during the training process in Figure 4. A clear widening gap emerges as $k$ increases, with RiskPO steadily surpassing GRPO across all evaluation points. This pattern indicates that the model is not merely improving its sampling efficiency on problems it could already solve, such as turning a "one success

Table 2: Pass@1 results on easy-level math benchmarks and multi-modal/coding benchmarks.

| Method | Easy-level math. reasoning | | | Multi-modal & coding | | |
|---|---|---|---|---|---|---|
| | MATH | GSM8K | Avg. | LCB | Geo3K | Avg. |
| GRPO | 54.3 | 78.8 | 66.55 | 25.8 | 53.7 | 39.75 |
| DAPO | 55.2 | 80.3 | 67.75 | 26.2 | 54.3 | 40.25 |
| RiskPO (Ours) | **56.2** | **80.3** | **68.25** | **26.8** | **54.5** | **40.65** |

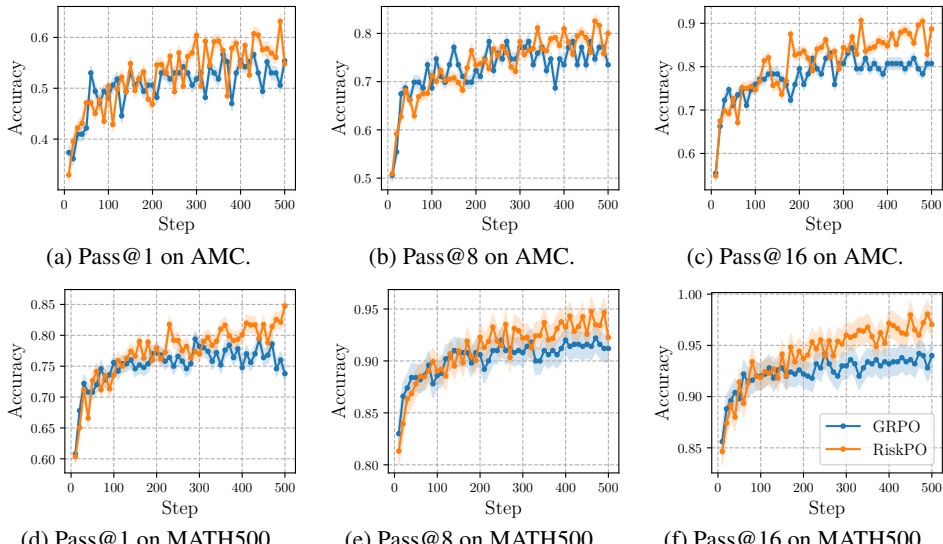

(a) Pass@1 on AMC.     (b) Pass@8 on AMC.     (c) Pass@16 on AMC.

(d) Pass@1 on MATH500.     (e) Pass@8 on MATH500.     (f) Pass@16 on MATH500.

Figure 4: Pass@k learning curves on the AMC and MATH500 datasets.

in sixteen attempts" case into a reliable single-shot success, but is in fact acquiring genuinely new solution strategies. Notably, RiskPO is able to solve instances that remain persistently unsolved under GRPO, even after sixteen attempts, all within the same computational budget. These results substantiate our claim that RiskPO expands the reasoning boundary of the base model, enabling progress beyond the capabilities achievable by conventional mean-based objectives.

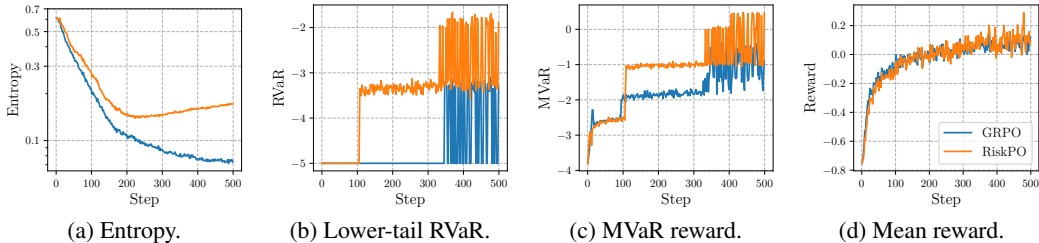

(a) Entropy.     (b) Lower-tail RVaR.     (c) MVaR reward.     (d) Mean reward.

Figure 5: Learning curves on DAPOMATH-17K, the RiskPO mitigates the entropy collapse and shows better performance on difficult problems, which is indicated by risk measures.

We investigate how the training dynamics of the RiskPO differ from GRPO, and Figure 5 depicts the trajectories of different objectives and entropy during training on the DAPOMATH-17K dataset. We contend that the mean reward is an inadequate training objective. The mean reward learning curves of GRPO and RiskPO are almost indistinguishable, with RiskPO exhibiting slightly greater fluctuations, likely due to its inherently higher-entropy behavior. In contrast, the lower-tail RVaR $\mathcal{J}_{\mathrm{RVaR}_{0:\alpha}}(\theta)$ and MVaR curves demonstrate a pronounced advantage for RiskPO. Since these risk-sensitive measures emphasize the lower tail of the reward distribution, higher values indicate stronger performance on the more challenging problems. Consistently, RiskPO maintains substantially higher entropy throughout training, whereas GRPO's entropy collapses early on, curtailing exploration and limiting its ability to tackle difficult instances.

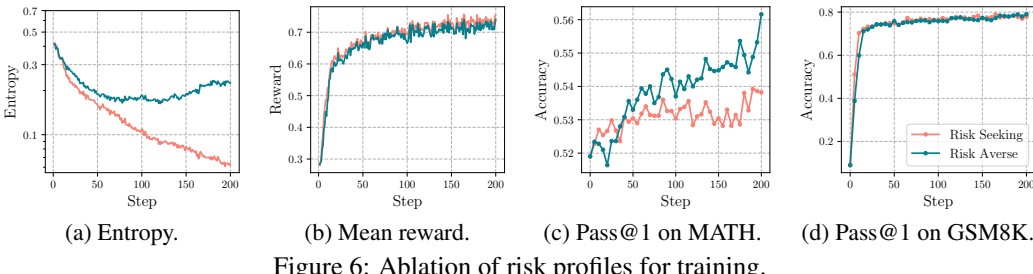

Figure 6: Ablation of risk profiles for training.

Table 3: Ablation of different quantile levels $(\alpha, \beta)$ on easy-level mathematical reasoning.

| Levels | $(\mathbf{0.2}, \mathbf{0.8})$ | $(\mathbf{0.1}, 0.8)$ | $(\mathbf{0.3}, 0.8)$ | $(\mathbf{0.4}, 0.8)$ | $(0.2, \mathbf{0.6})$ | $(0.2, \mathbf{0.7})$ | $(0.2, \mathbf{0.9})$ |
|---|---|---|---|---|---|---|---|
| **MATH** | **56.2** | 55.1 | 55.8 | 55.6 | 55.5 | 56.0 | 55.0 |
| **GSM8K** | **80.3** | 78.7 | 79.2 | 79.6 | 79.0 | 79.5 | 78.9 |
| **Avg.** | **68.25** | 66.90 | 67.50 | 67.60 | 67.25 | 67.75 | 66.95 |

## 6.2 ABLATION STUDY

We conduct the ablation study on the easy-level mathematics reasoning tasks. We start our analysis with the contrasting version:risk-seeking. As indicated by Section 5, focusing on the upper tail of the reward distribution will catalyse the entropy collapse. Similar to the MVaR objective, we use the counterpart risk-seeking objective, $(1 + \omega)(1 - \beta)\mathcal{J}_{\mathrm{RVaR}_{\beta:1}}(\theta) + (\beta - \alpha)\mathcal{J}_{\mathrm{RVaR}_{\alpha:\beta}}(\theta)$, to train the model and keep other parameters the same as the risk-averse version. The training curves are shown in Figure 6. The entropy of the risk-seeking version decreases sharply as the training proceeds, whereas the entropy of the risk-averse version decreases at the beginning, then remains stable around 0.2. In the Pass@1 curve on MATH, the risk-averse version exhibits a clear advantage over the risk-seeking. Before 50 steps, the risk-seeking has a better Pass@1 value. However, after the 50th step, the risk-seeking struggles to keep improving because it fails to optimize on those difficult problems (from 52% to 54%). The risk-averse version continues to improve on the Pass@1 (from 52% to 56%), showing 1.5 times improvement. This observation further justifies our theoretical results, which suggest the risk-averse objective is better than both the mean and risk-seeking objectives.

We further conduct a systematic investigation of the parameterization of the risk-averse algorithm, focusing on the quantile level $(\alpha, \beta)$. In the main experiments, we adopt $(0.2, 0.8)$ as the default configuration and independently perturb $\alpha$ and $\beta$ to validate this choice. The results of the quantile-level ablation are summarized in Table 3. Deviations from the configuration $(0.2, 0.8)$ consistently lead to a deterioration in performance. In particular, the configurations $(0.1, 0.8)$ and $(0.2, 0.9)$ exhibit more degradation. Reducing $\alpha$ to $0.1$ implies a diminished emphasis on the lower tail, whereas increasing $\beta$ to $0.9$ intensifies the emphasis on the upper tail. These observations underscore the importance of maintaining a risk-averse objective.

Table 4: Ablation of different bundle size $B$ on easy-level mathematical reasoning.

| Bundle size | 1 | 2 | 3 | 4 | **5** | 6 | 7 | 8 | 9 | 10 |
|---|---|---|---|---|---|---|---|---|---|---|
| **MATH** | 53.6 | 55.4 | 55.8 | 56.0 | **56.2** | 55.8 | 55.8 | 55.2 | 54.9 | 54.7 |
| **GSM8K** | 78.0 | 79.0 | 79.5 | 80.1 | **80.3** | 79.9 | 79.4 | 78.8 | 78.6 | 78.6 |
| **Avg.** | 65.80 | 67.20 | 67.65 | 68.05 | **68.25** | 67.85 | 67.6 | 67.00 | 66.75 | 66.65 |

We also report the ablation results on the bundle size in Table 4. The best performance is achieved at a bundle size of $B = 5$. This result can be intuitively explained by the trade-off in gradient estimation. Since all instances in a bundle share the same advantage estimation, using an overly large bundle dilutes the gradient signal, as too many instances rely on a single advantage value. Conversely, when the bundle size is too small, quantile tracking becomes unstable, which also weakens the gradient signal. The results in Table 4 corroborate this intuition: at $B = 2$ and $B = 10$, the average performance drops by $1.05\%$ and $1.6\%$, respectively. Setting $B = 1$ (no bundling) leads to the most severe degradation, with a $2.45\%$ drop. These findings highlight the necessity of bundling for RiskPO and suggest that bundle size should be carefully tuned, as both overly large and overly small values harm performance.

# 7 CONCLUSIONS

In this paper, we introduced RiskPO, a distributional alternative to mean-based objectives for reinforcement learning with verifiable reward. By leveraging MVaR and bundling multiple questions into informative training units, RiskPO effectively mitigates entropy collapse and strengthens exploration. Our theoretical analysis establishes that risk-averse updates weaken the coupling between policy log-probabilities and advantages, thereby preventing premature overconfidence. Empirically, RiskPO achieves consistent and significant improvements across a wide range of mathematical reasoning, code generation, and multi-modal benchmarks, outperforming GRPO and its strongest variants. These findings highlight that risk-averse objectives not only improve sample efficiency but also expand the reasoning frontier of LLMs.

## ACKNOWLEDGMENTS

This work was supported in part by the National Natural Science Foundation of China (NSFC) under Grants 72325007, and 72250065, and the Science and Technology Innovation Program of Hunan Province under Grant 2024RC7003.

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

## A  Experimental Setup and Supplementary Results

### A.1  Experimental Setup

**Model.**  We focus on mathematics reasoning, code generation, and multi-modal reasoning. We use DeepSeek-R1-Distill-Qwen-1.5B (Guo et al., 2025) as our base model to evaluate different algorithms on hard-level mathematics reasoning and code generation. On easy-level mathematics reasoning, we use Qwen2.5-Math-1.5B-Instruct (Yang et al., 2024) as the base model. On multi-modal reasoning, we use Qwen2.5-VL-Instruct-3B (Bai et al., 2025) as the base model.

**Training.**  For the hard-level mathematics reasoning tasks. We use the DAPO-math-17k as the training set. For the easy-level, we use MATH (Hendrycks et al., 2021) and GSM8K (Cobbe et al., 2021). For multi-modal reasoning, we use Geometry3K (Geo3K, Lu et al., 2021). For code generation, we train the models on Archer-6K (Wang et al., 2025a). We set the clipping threshold $\epsilon = 0.2$. KL penalty and entropy regularization are omitted from the loss objective. We use vLLM as the inference backend and FSDP as the training backend. We set the temperature to $0.8$ and top_p to $1.0$, and maximum output length as 3072. We generate 10 responses for each problem. The batch size is 512, the mini-batch size is set to 128. For quantile levels, we set $\alpha$ to $0.2$ and $\beta$ to $0.8$ correspondingly. The bundle size $B$ is set to 5. The mixing parameter of MVaR is $\omega = 0.5$. All training procedures are carried out on a Linux server equipped with 8 NVIDIA H20 GPUs, each providing 96 GB of memory.

**Evaluation.**  For hard-level mathematics reasoning, We evaluate on six math reasoning datasets: AIME24 (MAA, 2024) and AIME25 (MAA, 2025) with 30 problems from the American Invitational Mathematics Examination, both targeting advanced pre-collegiate reasoning; AMC23 (MAA, 2023) with 83 problems from the American Mathematics Competitions, testing creative algebraic, geometric, and number-theoretic skills; MATH-500 (Lightman et al., 2023) with 500 graduate-level problems from the original MATH dataset covering algebra, geometry, and number theory; Minerva Math (Lewkowycz et al., 2022) with 272 undergraduate-level quantitative reasoning problems; and OlympiadBench (He et al., 2024) with 675 Olympiad-style problems. For easy-level math tasks and the multi-modal task, we follow the train-test split in the original datasets. For code generations, we evaluate trained models on LiveCodeBench v5 (LCB, Jain et al., 2024).

### A.2  Ablation on the Mixing Parameter

We fix the quantile level, and perturb the $\omega$ to investigate how different attention on the lower tail influences the performance. The ablation of $\omega$ is shown in Table 5. Setting $\omega = 0.0$ would reduce the MVaR objective, $\mathcal{J}_{\mathrm{MVaR}^{\omega}_{\alpha:\beta}}(\theta)$, to $\mathcal{J}_{\mathrm{RVaR}_{0:\beta}}(\theta)$, which does not have extra attention on the lower tail even though it is still a risk-averse objective. The variant with $\omega = 0.0$ has the largest performance decrease, indicating the significance of extra attention on the lower tail. When setting $\omega = 1.0$, the variant also suffers from a mild performance drop. Overall, the phenomena suggest that the level of risk aversion needs to be properly tuned; both an indifferent level and excessive focus would lead to undesirable performance.

Table 5: Ablation of the mixing parameter.

| Settings | 0.0 | 0.1 | 0.5 | 0.6 | 1.0 |
|---|---|---|---|---|---|
| **MATH** | 54.8 | 55.1 | 56.2 | 56.0 | 55.7 |
| **GSM8K** | 79.6 | 80.0 | 80.3 | 80.2 | 80.0 |
| **Avg.** | 67.20 | 67.55 | 68.25 | 68.10 | 67.85 |

### A.3  Extensive Avg@k and Pass@k Results

Since both AIME2024 and AIME2025 contain only 30 questions, the Pass@k metric exhibits high variance and fluctuates significantly. To obtain a more stable evaluation, we report the Avg@k results in Figure 7. Across both datasets and different $k$ values, RiskPO consistently outperforms

GRPO, achieving higher Avg@k scores and demonstrating more stable improvements during training. The advantage of RiskPO is especially pronounced in the later training stages, where it continues to increase while GRPO tends to plateau. These results further confirm the effectiveness of our risk-sensitive optimization in enhancing reasoning performance on small-scale but challenging benchmarks like AIME.

Figure 8 reports Pass@k for $k \in \{1, 8, 16\}$. Across both datasets and all $k$, RiskPO consistently outperforms GRPO throughout training. The margin is modest but stable at $k = 1$ (especially on MINERVA, where variance is higher), and becomes clearly larger for $k = 8, 16$. This widening gap at larger $k$ indicates that RiskPO not only improves the best single prediction, but also spreads probability mass over a broader set of valid solution paths, thereby increasing the likelihood that at least one sampled response is correct. In effect, the risk-sensitive objective enhances coverage and diversity of reasoning, pushing the success frontier on problems that initially have low correctness probability and yielding larger gains at higher $k$.

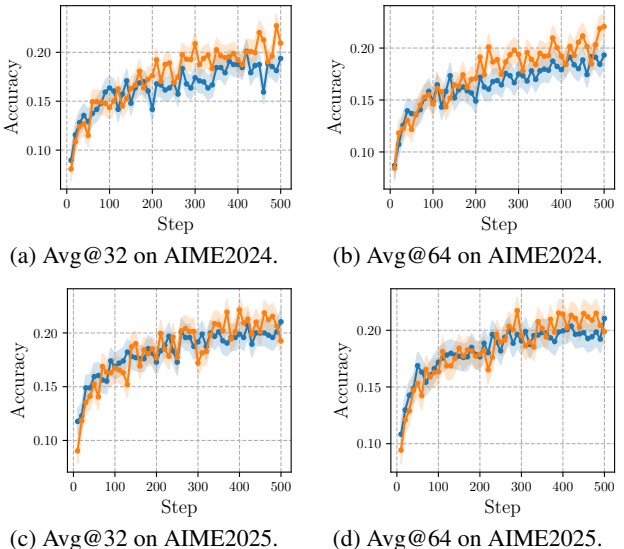

(a) Avg@32 on AIME2024.  (b) Avg@64 on AIME2024.

(c) Avg@32 on AIME2025.  (d) Avg@64 on AIME2025.

Figure 7: Avg@k learning curves on AIME2024 and AIME2025 datasets.

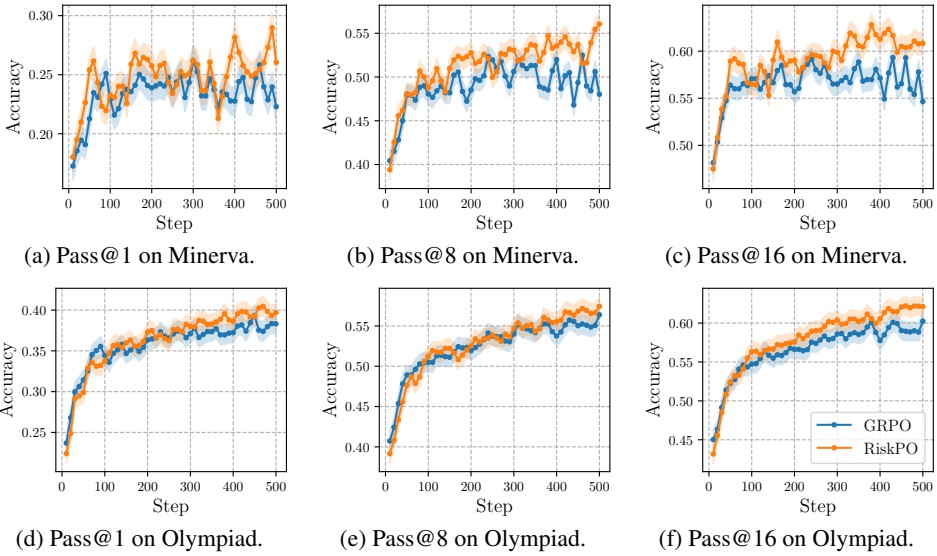

(a) Pass@1 on Minerva.  (b) Pass@8 on Minerva.  (c) Pass@16 on Minerva.

(d) Pass@1 on Olympiad.  (e) Pass@8 on Olympiad.  (f) Pass@16 on Olympiad.

Figure 8: Pass@k learning curves on Minerva and Olympiad datasets.

## A.4 DYNAMICS OF COVARIANCE AND ENTROPY

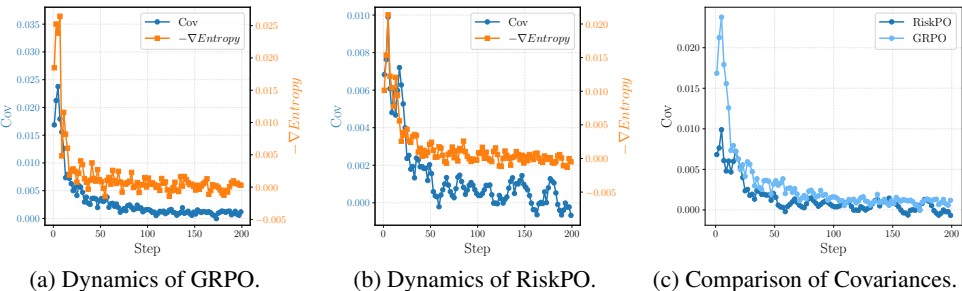

(a) Dynamics of GRPO.  (b) Dynamics of RiskPO.  (c) Comparison of Covariances.

Figure 9: The dynamics of the covariance and the entropy difference. We calculate the one step entropy difference and the covariance between log-prob and advantage.

In the Figure 9, we verify the validity of the Proposition 1 and Theorem 2. We record the covariance between log-probability and the advantage, and calculate the entropy difference during the training on easy-level math task. Figure 9a and 9b shows that the covariance and the entropy difference move in synchronicity during both the GRPO and RiskPO training, which validate our Proposition 1. To validate the Theorem 2, we show the comparison of covariance between RiskPO and GRPO in Figure 9c. The covariance of RiskPO is consistently smaller then the GRPO throughout the training, which coincides with the conclusion in the Theorem 2.

## A.5 JUSTIFICATION OF ASSUMPTION 1

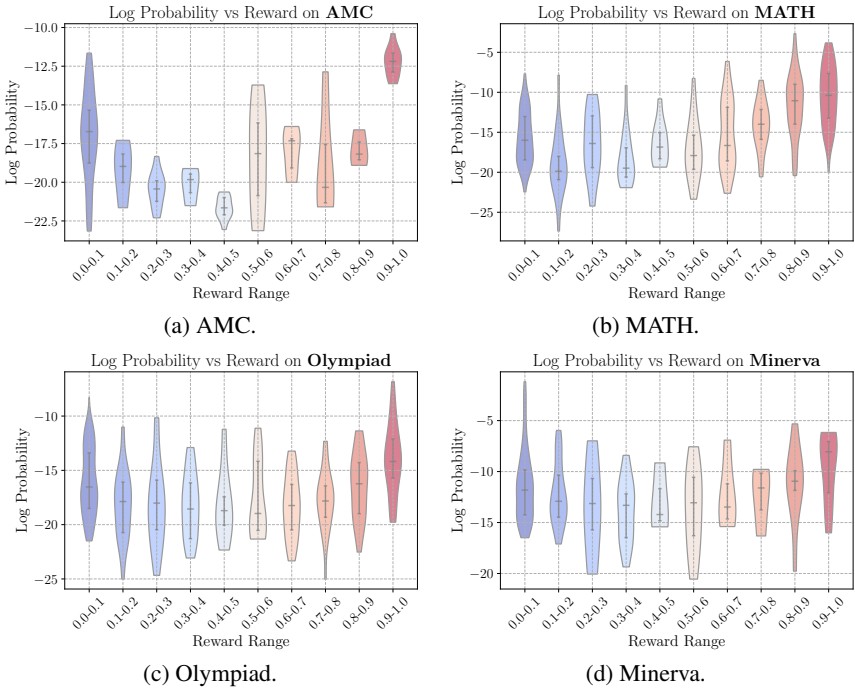

(a) AMC.  (b) MATH.

(c) Olympiad.  (d) Minerva.

Figure 10: The output log probability on various evaluation datasets.

Figure 10 presents the output log probability stratified by reward ranges across evaluation datasets. On Minerva and Olympiad datasets, the patterns closely align with Assumption 1: the output log probability is monotone with respect to reward in both the lower- and upper-tail regions, approximately on $(0, 0.3)$ and $(0.7, 1)$. Results on AMC show a similar monotone trend, although fluctuations appear in the mid-reward ranges, suggesting mixed difficulty and solution modes. For MATH,

which is comparatively easier, the upper tail exhibits strong monotonicity, while the lower tail is less pronounced—likely due to a scarcity of truly difficult items that would populate that region. Importantly, these evaluation sets are not used for model training; the observed regularities therefore provide additional evidence that Assumption 1 holds broadly for the pretrained base model across diverse benchmarks.

## B  THEORETICAL DETAILS

### B.1  PROOF OF THEOREM 1

*Proof.* Recall the definition of the RVaR functional:

$$\mathcal{J}_{\text{RVaR}_{\alpha:\beta}}(\theta) = \mathbb{E}\big[R(y)|R(y) \in [F_\theta^{-1}(\alpha), F_\theta^{-1}(\beta)]\big] = \frac{1}{\beta - \alpha} \int_{F_\theta^{-1}(\alpha)}^{F_\theta^{-1}(\beta)} r f_\theta(r) dr.$$

To compute the RVaR gradient, we apply Leibniz's rule for differentiation, yielding

$$\nabla_\theta \mathcal{J}_{\text{RVaR}_{\alpha:\beta}}(\theta) = \frac{1}{\beta - \alpha} \left( \int_{F_\theta^{-1}(\alpha)}^{F_\theta^{-1}(\beta)} r \nabla_\theta f_\theta(r) dr + F_\theta^{-1}(z) f_\theta(F_\theta^{-1}(z)) \nabla_\theta F_\theta^{-1}(z) \Big|_\alpha^\beta \right).$$

Note that, by the implicit function theorem, the quantile gradient can be expressed as (see, e.g., Fu et al., 2009) $\nabla_\theta F_\theta^{-1}(z) = -\nabla_\theta F_{\bar\theta}(F_\theta^{-1}(z))|_{\bar\theta=\theta} / f_\theta(F_\theta^{-1}(z))$. Substituting this identity into our previous expression, we can obtain

$$\nabla_\theta \mathcal{J}_{\text{RVaR}_{\alpha:\beta}}(\theta) = \frac{1}{\beta - \alpha} \left( \int_{F_\theta^{-1}(\alpha)}^{F_\theta^{-1}(\beta)} r \nabla_\theta f_\theta(r) dr - F_\theta^{-1}(z) \nabla_\theta F_{\bar\theta}(F_\theta^{-1}(z))|_{\bar\theta=\theta} \Big|_\alpha^\beta \right).$$

By the definition of CDF, we have $\nabla_\theta F_\theta(r) = \nabla_\theta \mathbb{E}\big[\mathbf{1}_{\{R(y) \le r\}}\big] = \mathbb{E}\big[\mathbf{1}_{\{R(y) \le r\}} \nabla_\theta \ln f_\theta(R(y))\big]$. Thus, with the score-function method, we rewrite the RVaR gradient in expectation form as

$$\nabla_\theta \mathcal{J}_{\text{RVaR}_{\alpha:\beta}}(\theta) = \mathbb{E}\left[ \left( R(y) \mathbf{1}_{\{R(y) \in [F_\theta^{-1}(\alpha), F_\theta^{-1}(\beta)]\}} - F_\theta^{-1}(z) \mathbf{1}_{\{R(y) \le F_\theta^{-1}(z)\}} \Big|_\alpha^\beta \right) \frac{\nabla_\theta \ln f_\theta(R(y))}{\beta - \alpha} \right].$$

Finally, since the distribution of $R(y)$ is induced by the LLM $\pi_\theta(\cdot|\cdot)$, we can apply the score-function transformation to yield the final expression

$$\nabla_\theta \mathcal{J}_{\text{RVaR}_{\alpha:\beta}}(\theta) = \frac{1}{\beta - \alpha} \mathbb{E}\big[g\big(R(y), F_\theta^{-1}(\alpha), F_\theta^{-1}(\beta)\big) \nabla_\theta \ln \pi_\theta(y|x)\big],$$

which completes the proof. $\qquad\square$

### B.2  PROOF OF PROPOSITION 1

*Proof.* With a Lipschitz-continuous entropy gradient and a bounded Hessian, the first-order Taylor expansion yields $\mathcal{H}(\pi_{\theta_{k+1}}|x) = \mathcal{H}(\pi_{\theta_k}|x) + \langle \nabla_\theta \mathcal{H}(\pi_{\theta_k}|x), \Delta_k \rangle + O(\|\Delta_k\|^2)$. The entropy gradient can be written as

$$\begin{aligned}
\nabla_\theta \mathcal{H}(\pi_\theta \,|\, x) &= \nabla_\theta(-\mathbb{E}_{y \sim \pi_\theta(\cdot|x)}[\log \pi_\theta(y|x)]) \\
&= -\mathbb{E}_{y \sim \pi_\theta(\cdot|x)}[\nabla_\theta \log \pi_\theta(y|x) + \log \pi_\theta(y|x) \nabla_\theta \log \pi_\theta(y|x)] \\
&= -\mathbb{E}_{y \sim \pi_\theta(\cdot|x)}[\log \pi_\theta(y|x) \nabla_\theta \log \pi_\theta(y|x)],
\end{aligned}$$

where the second equality comes from the score-function method, and the last equality is due to the identity $\mathbb{E}_{y \sim \pi_\theta(y|x)}[\nabla_\theta \log \pi_\theta(y|x)] = 0$. Note that $\frac{\partial}{\partial \theta_{x,y'}} \log \pi_\theta(y|x) = \mathbf{1}_{\{y=y'\}} - \pi_\theta(y'|x)$. Taking the inner product with $\Delta$ gives

$$\begin{aligned}
\langle \nabla_\theta \mathcal{H}(\pi_\theta|x), \Delta \rangle &= -\langle \mathbb{E}_{y \sim \pi_\theta(\cdot|x)}\big[\log \pi_\theta(y|x) \nabla_\theta \log \pi_\theta(y|x)\big], \Delta \rangle \\
&= -\mathbb{E}_{y \sim \pi_\theta(\cdot|x)}\big[\log \pi_\theta(y|x) \langle \nabla_\theta \log \pi_\theta(y|x), \Delta \rangle\big] \\
&= -\mathbb{E}_{y \sim \pi_\theta(\cdot|x)}\left[\log \pi_\theta(y|x) \sum_{y' \in \mathcal{Y}} \frac{\partial \log \pi_\theta(y|x)}{\partial \theta_{x,y'}} \Delta_{x,y'}\right] \\
&= -\mathbb{E}_{y \sim \pi_\theta(\cdot|x)}\left[\log \pi_\theta(y|x) \sum_{y' \in \mathcal{Y}} (\mathbf{1}_{\{y=y'\}} - \pi_\theta(y'|x)) \Delta_{x,y'}\right] \\
&= -\mathbb{E}_{y \sim \pi_\theta(\cdot|x)}\big[\log \pi_\theta(y|x) \Delta_{x,y}\big] - \mathbb{E}_{y \sim \pi_\theta(\cdot|x)}\big[\log \pi_\theta(y|x)\big] \sum_{y' \in \mathcal{Y}} \pi_\theta(y'|x) \Delta_{x,y'} \\
&= -\text{Cov}_{y \sim \pi_\theta(\cdot|x)}\big(\log \pi_\theta(y|x), \Delta_{x,y}\big) \qquad\qquad (6)
\end{aligned}$$

In a tabular softmax policy, a natural policy gradient update step admits $\Delta_{x,y} = \eta A_\theta(x,y)$ (Agarwal et al., 2021). Plugging this into (6) yields the equality (4), which completes the proof. $\qquad\square$

### B.3 PROOF OF THEOREM 2

Before proving Theorem 2, we first prepare the following lemma, which provides a convenient representation of covariance.

**Lemma 1.** *Let $X, Y$ be real-valued random variables satisfying $\mathbb{E}[|XY|] < \infty$. Then*

$$\mathrm{Cov}(X, Y) = \int_{-\infty}^{\infty} \mathrm{Cov}\big(\mathbf{1}_{\{X>t\}}, Y\big) \, dt. \tag{7}$$

*Proof.* We start from the layer–cake representation $X = \int_0^\infty \big(\mathbf{1}_{\{X>t\}} - \mathbf{1}_{\{-X>t\}}\big) \, dt$, which holds for any real-valued $X$. Multiplying by $Y$ and taking expectations, we may apply the Tonelli–Fubini theorem under the integrability condition $\mathbb{E}[|XY|] < \infty$, which yields

$$\mathbb{E}[XY] = \int_0^\infty \Big( \mathbb{E}\big[Y \, \mathbf{1}_{\{X>t\}}\big] - \mathbb{E}\big[Y \, \mathbf{1}_{\{-X>t\}}\big] \Big) \, dt.$$

Applying the same transformation to $\mathbb{E}[X]\mathbb{E}[Y]$ and subtracting, we obtain

$$\mathrm{Cov}(X, Y) = \int_0^\infty \Big( \mathrm{Cov}\big(\mathbf{1}_{\{X>t\}}, Y\big) - \mathrm{Cov}\big(\mathbf{1}_{\{-X>t\}}, Y\big) \Big) \, dt.$$

Changing the integral variable in the second term and using the identity $-\mathrm{Cov}(1 - Z, Y) = \mathrm{Cov}(Z, Y)$, we merge the two integrals and obtain the equality (7). The distinction between strict and non-strict inequalities only affects a countable set of $t$ values and does not change the integral under the Lebesgue measure, which completes the proof. $\square$

Next, we present the proof of Theorem 2. For notational clarity, we focus on a single representative output $y$ from the model $\pi_\theta(\cdot|x)$ rather than a bundle. The advantage values for the MVaR- and mean-based objectives are given by $A_{\mathrm{MVaR}_{\alpha:\beta}^\omega} = -(1 + \omega)(F_\theta^{-1}(\alpha) - R(y))^+ + g(R(y), F_\theta^{-1}(\alpha), F_\theta^{-1}(\beta))$ and $A_{\mathrm{Mean}} = R(y) - \mathbb{E}[R(y)]$.

*Proof of Theorem 2.* Recall that the positive part function can be expressed via the layer–cake representation: $(z - a)^+ = \int_a^{+\infty} \mathbf{1}_{\{z>t\}} \, dt$. With Lemma 1, we can derive the covariances as below

$$\mathrm{Cov}(A_{\mathrm{MVaR}_{\alpha:\beta}^\omega}, \mathrm{SF}) = \left[ (1 + \omega) \int_{-\infty}^{F^{-1}(\alpha)} + \int_{F^{-1}(\alpha)}^{F^{-1}(\beta)} \right] \mathrm{Cov}\big(\mathbf{1}_{\{R(y)>t\}}, \mathrm{SF}\big) \, dt, \tag{8}$$

where, for notational convenience, we denote $\mathrm{SF} := \log \pi_\theta(y|x)$. Define $k(t) := \mathrm{Cov}(\mathbf{1}_{\{R(y)>t\}}, \mathrm{SF})$ and recall that the density of $R(y)$ is $f_\theta$. Then, we compute the derivative of $k(t)$ as follows:

$$\begin{aligned}
k'(t) &= \frac{d(\mathbb{E}[\mathbf{1}_{\{R(y)>t\}}\mathrm{SF}])}{dt} - \frac{d(\mathrm{Pr}(R(y) > t)\mathbb{E}[\mathrm{SF}])}{dt} \\
&= \frac{d}{dt}\mathbb{E}\big[\mathbb{E}[\mathbf{1}_{\{R(y)>t\}}\mathrm{SF}|R(y)]\big] - \mathbb{E}[\mathrm{SF}]\frac{d}{dt} \int_t^\infty f_\theta(r) dr \\
&= \frac{d}{dt}\mathbb{E}\big[\mathbf{1}_{\{R(y)>t\}}\mathbb{E}[\mathrm{SF}|R(y)]\big] - \mathbb{E}[\mathrm{SF}]\frac{d}{dt} \int_t^\infty f_\theta(r) dr \\
&= \frac{d}{dt} \int_t^\infty \mathbb{E}[\mathrm{SF}|R(y) = r]f_\theta(r) dr - \mathbb{E}[\mathrm{SF}]\frac{d}{dt} \int_t^\infty f_\theta(r) dr \\
&= -\psi(t)f_\theta(t) - (-\mathbb{E}[\mathrm{SF}]f_\theta(t)) \\
&= -f_\theta(t)\big(\psi(t) - \mathbb{E}[\mathrm{SF}]\big).
\end{aligned}$$

Under Assumption 1, $\psi(t) \geq \mathbb{E}[\mathrm{SF}]$ for $t \geq F_\theta^{-1}(\beta)$ and $t \leq F_\theta^{-1}(\alpha)$. Consequently, $k'(t) \leq 0$ for $t \geq F_\theta^{-1}(\beta)$ and $t \leq F_\theta^{-1}(\alpha)$, which implies that $k(t)$ is non-increasing in both the upper and lower tails. Moreover, since $\mathbb{E}[|\mathrm{SF}|] < \infty$, the dominated convergence theorem implies that

$$\begin{aligned}
\lim_{t\to\infty} k(t) &= \lim_{t\to\infty} \mathrm{Cov}(\mathbf{1}_{\{R(y)>t\}}, \mathrm{SF}) \\
&= \lim_{t\to\infty} \mathbb{E}[\mathbf{1}_{\{R(y)>t\}}\mathrm{SF}] - \lim_{t\to\infty} \mathrm{Pr}(R(y) > t)\mathbb{E}[\mathrm{SF}] \\
&= \mathbb{E}[\lim_{t\to\infty} \mathbf{1}_{\{R(y)>t\}}\mathrm{SF}] - 0 = 0.
\end{aligned}$$

Analogously, we can show that $\lim_{t \to -\infty} k(t) = \mathbb{E}[\text{SF}] - \mathbb{E}[\text{SF}] = 0$. By the monotonicity in the tails, we obtain $k(t) \geq 0$ for $t \geq F_\theta^{-1}(\beta)$ and $k(t) \leq 0$ for $t \leq F_\theta^{-1}(\alpha)$. Therefore, noting that $k(t) = \text{Cov}\left(\mathbf{1}_{\{R(y)>t\}}, \text{SF}\right)$ preserves its sign on both tails, we can further obtain

$$\text{Cov}(A_{\text{MVaR}_{\alpha:\beta}^\omega}, \text{SF}) \leq \int_{\mathbb{R}} \text{Cov}\left(\mathbf{1}_{\{R(y)>t\}}, \text{SF}\right) dt = \text{Cov}(A_{\text{Mean}}, \text{SF}),$$

which completes the proof. $\qquad\square$

### B.4 SUPPLEMENTARY THEOREM OF SECTION 5

With the different treatment of the tail in the reward distribution, we obtain the following covariance result between the resulting advantage value and the output log-probability.

**Theorem 3.** *If Assumption 1 holds with $\mathbb{E}[|\text{SF}|] < \infty$, and $g_1, g_2$ are nondecreasing and differentiable with $g_1'(t) \geq g_2'(t)$ on $[F_\theta^{-1}(\beta), \infty)$, $g_1(t) = g_2(t)$ on $(-\infty, F_\theta^{-1}(\beta)]$, then we have*

$$\text{Cov}(\text{SF}, g_2(R(y))) \geq \text{Cov}(\text{SF}, g_1(R(y))).$$

*Proof.* Note that

$$\text{Cov}(\text{SF}, g_i(R(y))) = \int_{-\infty}^{\infty} \text{Cov}(\text{SF}, \mathbf{1}_{\{g_i(R(y))>t\}}) dt = \int_{-\infty}^{\infty} \text{Cov}(\text{SF}, \mathbf{1}_{\{R(y)>u\}}) dg_i(u)$$

$$= \int_{-\infty}^{\infty} \text{Cov}(\text{SF}, \mathbf{1}_{\{R(y)>t\}}) g_i'(t) dt,$$

where the first equality is due to Lemma 1 and the second equality is integration by substitution. The proof of Theorem 2 implies that under Assumption 1, we have $k(t) \geq 0$ for $t \geq F_\theta^{-1}(\beta)$, thus

$$\text{Cov}(\text{SF}, g_2(R(y))) - \text{Cov}(\text{SF}, g_1(R(y))) = \int_{F_\theta^{-1}(\beta)}^{\infty} k(t)(g_2'(t) - g_1'(t)) dt \geq 0,$$

which gives the desired result. $\qquad\square$

A symmetric conclusion can be derived analogously for the treatment of the other tail.

