# OpenReview forum: "RiskPO: Risk-based Policy Optimization with Verifiable Reward for LLM Post-Training"
_ICLR.cc/2026/Conference — ICLR 2026 Poster_

### Official Review · Reviewer_yqFg · 2025-10-31

**Soundness:** 3
**Presentation:** 3
**Contribution:** 2
**Rating:** 6
**Confidence:** 3

**Summary:**

The paper introduces Risk-based Policy Optimization (RiskPO), a post-training reinforcement learning (RL) framework for large language models (LLMs) that replaces the standard mean-based GRPO objective with a risk-sensitive objective called Mixed Value-at-Risk (MVaR).
The motivation is that GRPO’s mean-based objective focuses on high-probability “easy” reasoning trajectories, causing entropy collapse and limited exploration. RiskPO instead amplifies gradient signals from low-reward or rare reasoning paths, encouraging exploration of harder problems.
The authors bundle multiple questions to enrich the binary reward signal, analyze entropy theoretically, and show that the risk-averse formulation mitigates overconfidence and improves reasoning diversity.

Extensive experiments on over ten benchmarks—including AIME24/25, Minerva, Olympiad, MATH500, and AMC—show that RiskPO outperforms GRPO, DAPO, and Dr.GRPO, achieving higher Pass@k and Avg@k metrics. The improvement is more pronounced at higher k, indicating broader coverage of valid reasoning trajectories.

**Strengths:**

- Clearly identifies the limitations of mean-based GRPO objectives and motivates a principled alternative.
- Proposes a novel risk-sensitive objective (MVaR) that enhances exploration and mitigates entropy collapse.
- Shows consistent performance improvements across reasoning benchmarks, especially on harder problems.

**Weaknesses:**

- Experiments are mostly limited to mathematical reasoning tasks, with unclear generalization to other domains.
- The empirical analysis of how risk sensitivity affects exploration or diversity is insufficient.

**Questions:**

See weaknesses.

---

> ### Author Response · Authors · 2025-11-22
> **Rebuttal**
>
> > *[W1]. Experiments are mostly limited to mathematical reasoning tasks, with unclear generalization to other domains.*
>
> **RW1**
>
> We sincerely appreciate the reviewer’s forward-looking comments on generalization beyond our current RLVR setting. In this work, we propose a novel risk-sensitive policy-gradient algorithm and instantiate it for RLVR to improve LLM reasoning ability. While the main experiments focus on mathematical reasoning, we also include code generation and multimodal reasoning tasks, which are central application areas for LLM post-training and RLVR.
>
> From an algorithmic perspective, our MVaR-based RiskPO is a general distributional RL method that does not rely on any math-specific structure. In principle, it can be applied to classical RL benchmarks (e.g., Atari, MuJoCo), broader RLHF pipelines, and vision–language–action (VLA) or tool-use settings. Exploring these domains is a promising direction for future work. More broadly, any application that currently uses a mean-based policy-gradient update could, in principle, be enhanced by replacing it with our risk-sensitive RiskPO objective.
>
> > *[W2]. The empirical analysis of how risk sensitivity affects exploration or diversity is insufficient.*
>
> **RW2**
>
> We have conducted the ablation study on diferent risk level (or quantile level) in the Table 3. To further investigate how risk sensitivity affects exploration and the performance, we add the experiment to show how the model perform on easy and hard question. We use MATH500 and Olympiad datasets since the two datasets contain the most questions. For each questions, we require the model to generate 16 responses and rank the question according to the success rate. Then, we can partition the dataset into two parts with equal number of question; part one is the partition with easy question and part two is the difficult one. We evaluate the model's Pass@1 accuracy and the corresponding average policy entropy on different datasets (the entropy value is shown within the parentheses). The results are shown in the following Table. It shows that on easy question the RiskPO has comparable performance with the GRPO while maintaining sufficient entropy. On difficult questions, the RiskPO has clear advantage over GRPO, and GRPO suffers from entropy collapse which results in limited performance improvement.
>
> | method | MATH500 (part one) | MATH500 (part two) |
> |-|-|-|
> | base | 82.3+-2.6 (0.38) | 37.6+-2.5 (0.35) |
> | GRPO | 97.6+-1.2 (0.08) | 60.8+-1.8 (0.16) |
> | RiskPO | 97.8+-1.2 (0.13) | 65.8+-1.4 (0.22) |
>
> | method | Olympiad (part one) | Olympiad (part two) |
> |-|-|-|
> | base | 42.8+-2.5 (0.29) | 18.3+-2.3 (0.32) |
> | GRPO | 54.4+-1.1 (0.04) | 25.3+-1.8 (0.10) |
> | RiskPO | 54.4+-1.2 (0.13) | 28.6+-2.1 (0.18) |

---

### Official Review · Reviewer_Hp4w · 2025-11-01

**Soundness:** 2
**Presentation:** 2
**Contribution:** 2
**Rating:** 4
**Confidence:** 4

**Summary:**

This paper introduces RiskPO, a reinforcement learning algorithm method for LLM post-training. RiskPO replaces the original objective in GRPO with a risk-sensitive objective called Mixed Value-at-Risk (MVaR). The key motivation is that methods like GRPO suffer from entropy collapse and poor exploration. RiskPO uses bundled questions and tail-focused optimization to amplify gradient signals from rare but informative low-reward cases, thereby promoting exploration and better reasoning. Experiments show that RiskPO outperforms other baselines on several benchmarks and mitigates the overconfidence issue.

**Strengths:**

- This paper investigates a question of significant concern to the community: the entropy collapse of GRPO during training and the sparse reward issue on difficult problems.
- It’s very interesting to incorporate a risk-sensitive objective based on the Mixed Value-at-Risk into the advantage estimation. The detailed theoretical explanation makes it clearer.
- RiskPO shows promising performance gain compared to other baselines. The ablation study is thorough.

**Weaknesses:**

- The motivation is not convincing enough. The authors argue that GRPO suffers from entropy collapse, which is indeed a consensus in the community. However, they attribute this to overemphasizing high-probability output without any experimental or theoretical verification. Due to the design of the GRPO advantage, the model's gradient is 0 on both all-correct and all-incorrect problems, only updating samples with performance differences within the same group. I'd like to know how the authors correlate these non-zero advantage samples with high probability; is there any experimental verification or theoretical derivation?
- I think considering GRPO as a mean-based method is debatable, since there is a significant difference between directly using reward signals for gradient optimization and using advantages. Therefore, I do not consider GRPO to be truly maximizing the expected average reward. In my view, RiskPO essentially extends the original group definition in GRPO, using different baseline definitions to mitigate the sparse reward problem in difficult tasks. In this context, RiskPO and GRPO should be classified similarly in terms of their objectives.
- Although the proposed bundling strategy is intended to approximate a continuous reward distribution, the underlying feedback remains binary in nature. This discreteness introduces a potential mismatch between the analytical assumptions in Section 4 (which rely on smooth reward distributions) and the practical training setup. Moreover, the quality of the MVaR-based advantage estimation may depend sensitively on the bundle and generation sizes (B and G). Given that both are relatively small in the experiments, it remains unclear whether the estimated quantiles accurately reflect the model’s true reward distribution at each training step. Additional evidence or analysis would strengthen the authors’ claim that MVaR optimization remains valid under such discrete and limited sampling conditions.
- The experimental results should be expanded, e.g., on larger models. Additionally, the settings of the current experiments should be described more clearly. For Tables 1 and 2, how many samples were used to compute Pass@1? Some baselines perform worse than vanilla GRPO, and I wonder why. Disclosing this information would make the results more convincing.

**Questions:**

- Can you further explain why the bundle grouping makes sense?
- What does “papers” refer to in lines 251 and 252?
- I think the connection between Sec. 5 and Secs. 3,4 is somewhat weak. Could you further explain why strengthening the gradients on hard questions can address the issue of overconfident convergence?
- Please see Weaknesses for the other questions mentioned.

---

> ### Author Response · Authors · 2025-11-22
> **Rebuttal part 1**
>
> > *[W1]. The motivation is not convincing enough.*
>
> **RW1**
> Thank you for raising concerns about the motivation. We establish **the theoretical analysis supported by empirical observation in section 5** to demonstrate why GRPO cause entropy collapse and the RiskPO can mitigate the dilemma. First, we give the **per-step entropy analysis in Proposition 1**, which indicates that the decrease of entropy is proportional to the covariane between log-probability and the advantage. Then, we **made an assumption that the log-prob has monotonicity** in upper and lower reward areas. The assumption can be verified by empirical evidence shown in Figure 3 and Figure 9. Finally, we can **derive Theorem 2 to show that the RisPO has lower covariance then the GRPO,** thus the entropy collapse can be mitigated by using RiskPO.
>
> The proof of Theorem 2 proceeds by explicitly **decomposing the covariance** between the log-probability and the advantage into contributions from different reward regions. **GRPO uses a mean-based advantage that weights all rewards equally,** so high-reward (and thus typically high-probability) trajectories contribute excessively to this covariance. Our MVaR-based advantage, **in contrast, downweights the upper tail and emphasizes the lower tail of the reward distribution.** Under Assumption 1, this reweighting directly reduces the covariance between log-probability and advantage, which is exactly the quantity governing entropy decay in Proposition 1. Intuitively, GRPO “averages” over all problems, while RiskPO redistributes this budget toward the more difficult tail. We further corroborate this mechanism with a risk-seeking variant in our experiments: when we deliberately upweight the upper reward tail, the covariance term increases and we observe even faster entropy collapse, providing a complementary sanity check for our theoretical explanation.
>
> **The over-emphasis happens in the transition regime where both correct and incorrect responses exist.** In practice, the dominant part of training does not lie at the all-wrong and all-correct regimes (zero gradient), but in the transitional regime where a question is solved with non-trivial probability (some samples correct, some incorrect). In this regime, because higher-reward responses tend to have higher log-probabilities (Assumption 1 and Fig. 3), the non-zero GRPO advantages are positively correlated with log-probability. Proposition 1 then implies that these updates systematically decrease entropy and reinforce high-probability trajectories, which is what we mean by “over-emphasizing easy problems.”
>
> By contrast, the zero-gradient effect that actually hurts performance is the all-wrong case, not the all-correct case. Once a problem has become trivially easy (all-correct), it is acceptable that GRPO stops updating it. However, truly difficult problems often remain in the all-wrong regime for a long time: the standardized advantage is 0, and these questions receive essentially no learning signal throughout training. This is precisely the failure mode we target with our bundle-wise, risk-averse design: by aggregating multiple questions and defining an MVaR-based advantage on the bundle score, hard questions contribute to non-zero bundle advantages even when they are individually always wrong, so they do receive gradient and can be improved.
>
> Finally, we note that optimizing the mean reward from the feedback on a single question is itself only a proxy for the true RLVR goal, namely improving test-time metrics such as Pass@k on held-out problems. Our MVaR + bundling design can be viewed as a strict generalization of existing mean-based GRPO-style objectives: when $\alpha=0, \beta=1$, and the bundle size $B=1$, RiskPO reduces exactly to the standard formulation. Thus, in principle, RiskPO covers the original objective both in form and scale, while allowing strictly richer risk-sensitive behavior, and can match or improve upon GRPO-type methods under appropriate hyperparameter choices.

---

> ### Author Response · Authors · 2025-11-22
> **Rebuttal part 2**
>
> > *[W2]. I think considering GRPO as a mean-based method is debatable.*
>
> **RW2**
> We appreciate the reviewer for pointing out that our use of the term mean-based may be ambiguous. In this paper, when we refer to “mean-based methods” we specifically mean algorithms whose **underlying risk functional is the expected reward (or any affine transformation of it)**, rather than methods that literally plug raw rewards into the gradient. In standard policy-gradient theory, **replacing rewards by advantages is a classical variance-reduction technique** and does not change the fact that, in expectation, the algorithm is optimizing the mean return.
>
> In GRPO, the group-wise standardized advantage is defined as $A_i=\frac{R(y_i)-\bar R}{\text{std}}$, where $\bar R$ and $\text{std}$ are the sample mean and standard deviation with the group. This is an affine transformation of the sequence-level reward. The policy gradient takes the form $1/\text{std}*\mathbb{E}[R(y)\nabla_\theta \log \pi_\theta(y\vert x)-\bar R \nabla_\theta \log \pi_\theta(y\vert x)]$. The second term vanishes by the zero-mean property of score function $\mathbb{E}[\nabla_\theta \log \pi_\theta(y\vert x)]$, so the expected gradient is proportional to $\mathbb{E}[R(y) \nabla_\theta \log \pi_\theta(y\vert x)]$, i.e., the gradient of the expected reward. Although GRPO uses a group-wise advantage instead of raw rewards, its asymptotic objective is still to maximize average reward; this is precisely what we mean when we categorize GRPO as a mean-based method.
>
> By contrast, RiskPO differs from GRPO not because it also uses a different advantage, but because this advantage comes from a non-linear risk functional of the entire reward distribution. We first aggregate per-question rewards into a bundle score $R_B=\sum_{i=1}^BR(y_i)$ and then optimize the Mixed-Valued-at-Risk (MVaR) objective, $J^{MVaR}_{\alpha:\beta,\omega}(\theta)=(1+\omega)\int^{F^{-1}\_{\theta}(\alpha)}\_{F^{-1}\_{\theta}(0)}rdF\_\theta(r)+\int^{F^{-1}\_{\theta}(\beta)}\_{F^{-1}\_{\theta}(\alpha)}rdF\_\theta(r)$, where F is the CDF of the bundle-wise score. The MVaR objective puts extra weight on the lower tail of the bundle-score distribution while excluding the highest-performing areas. This functional cannot be reduced to the mean $\mathbb{E}[R_B]$ by any affine transformation, except in the degenerate case $(\alpha, \beta)=(0,1), \omega=0$, where MVaR collapses to the expectation. Thus even though we implement RiskPO in a policy-gradient form, the algorithm maximize a risk-averse objective, not the mean.
>
> We also agree that the notions of “group” in GRPO and “bundle” in RiskPO may look similar at first glance, but their roles are fundamentally different. In GRPO, a group of $G$ responses to the same question serves as a Monte Carlo baseline to reduce variance for gradient estimation. In RiskPO, we still draw $G$ responses per question, but we additionally form **bundles of $B$ different questions** and define the "paper score" as the sum of one sampled response from each question in the bundle (sampled without replacement across the bundle). The advantage is then computed at the bundle level, based on the distribution of these bundle scores under the MVaR criterion. This bundling mechanism does not introduce extra inference cost, but it enriches the effective reward signal by moving from binary per-question rewards to a more informative distribution over bundle scores. This is what allows RiskPO to revisit RLVR from a genuinely distributional optimization viewpoint, and in our ablations we explicitly isolate and quantify the gains brought by the bundling design.

---

> ### Author Response · Authors · 2025-11-22
> **Rebuttal part 3**
>
> > *[W3] and [Q1]. Although the proposed bundling strategy is intended to approximate a continuous reward distribution, the underlying feedback remains binary in nature. we treat each bundle as an atomic training unit with a multi-level return, and apply MVaR to the empirical distribution of these bundle scores. Across many bundles and training iterations, the empirical distribution of $R_B$ is richer and much closer to the “continuous” ideal assumed in the analysis than the raw per-question 0/1 feedback.
>
> **RW3 and R1**
> We thank the reviewer for pointing out this subtlety. In Section 4, we assume that the return distribution admits a density purely to obtain a compact integral expression for the MVaR objective and its gradient. **This is a standard analytical simplification in risk-sensitive reinforcement learning and stochastic approximation [1,2],** rather than a requirement of the algorithm itself. From a stochastic approximation viewpoint, our quantile trackers and policy parameters are updated by Robbins–Monro–type recursions that seek to solve the MVaR optimality condition for the actual return distribution induced by the policy [3]. **Classical results guarantee that such recursions converge to the solution of the underlying ODE or differential inclusion under mild regularity and step-size conditions, regardless of whether the return distribution is continuous or discrete [4].** In this sense, the continuous “density” case analyzed in Section 4 should be viewed as an idealized limit that makes the formula apparent, while the discrete bundle-return setting used in practice is its natural approximation: the same update rule remains stable and converges to essentially the same risk-sensitive region of policies, rather than producing qualitatively different or pathological behavior.
>
> In our setting, the basic optimization object is not a single 0/1 question, but the bundle score: $R_B=\sum_{i=1}^BR(y_i)$, whose support is ${0,\dots,B}$ instead of ${0,1}$. This is analogous to standard RL, where the natural training unit is can be an episode return rather than a single time step. The rationale for combining different questions into a bundle is that the solutions to some math and coding problems share similar intuitions. Combining them not only enriches the distributional information but also helps models learn by leveraging these shared reasoning pathways. We also believe that this problem formulation facilitates further exploration for long-CoT reasoning. We treat each bundle as an atomic training unit with a multi-level return, and apply MVaR to the empirical distribution of these bundle scores. Across many bundles and training iterations, the empirical distribution of $R_B$ is richer and much closer to the “continuous” ideal assumed in the analysis than the raw per-question 0/1 feedback. And we validate the necessity on the bundling in the Table 4, the MVaR without bundling performs the worst. We further conduct an ablation study to demonstrate that bundling is an indispensable component for algorithms like RiskPO, which are designed from a distributional perspective. As shown in the following table, we investigate the performance of mean-based GRPO with bundling. The results indicate that GRPO does not benefit significantly from bundling, as it only focuses on partial information (i.e., the mean) of the distribution. In contrast, our risk-sensitive method (RiskPO), which requires more comprehensive distributional information, benefits substantially from bundling.
>
> | method | GSM8K-Platinum | MATH |
> |-|-|-|
> | Mean(vanilla GRPO) | 78.9+-1.2 | 54.5+-1.2 |
> | Mean + Bundling | 79.0+-1.1 | 54.6+-1.3 |
> | CVaR-only + Bundling | 80.7+-0.9 | 56.0+-1.1 |
> | MVaR without bundling | 78.6+-1.2 | 53.8+-1.3 |
> | MVaR with bundling(RiskPO) | 82.6+-1.2 | 56.8+-0.8 |
>
> [1] Rowland, Mark, et al. "An analysis of quantile temporal-difference learning." Journal of Machine Learning Research 25.163 (2024): 1-47.
>
> [2] Tamar, Aviv, Yonatan Glassner, and Shie Mannor. "Optimizing the CVaR via sampling." Proceedings of the AAAI Conference on Artificial Intelligence. Vol. 29. No. 1. 2015.
>
> [3] Borkar, Vivek S., and Vivek S. Borkar. Stochastic approximation: a dynamical systems viewpoint. Vol. 100. Cambridge: Cambridge University Press, 2008.
>
> [4] Borkar, Vivek S., and Sean P. Meyn. "The ODE method for convergence of stochastic approximation and reinforcement learning." SIAM Journal on Control and Optimization 38.2 (2000): 447-469.

---

> ### Author Response · Authors · 2025-11-22
> **Rebuttal part 4**
>
> > *[W3]. The experimental results should be expanded, e.g., on larger models.*
>
>
> **RW4**
> Due to the limited computation resource, we did not present the results on larger models in the original submission. Here we add the results on DeepSeek-R1-Distill-Qwen-7B trained by GRPO and RiskPO for 200 steps. We present the Pass@1 accuracy with 95% confidence intervals in the ± half-width format. As shown in the following table, the RiskPO consistently out-performs GRPO.
>
> | method | AMC | MATH500 | Minerva | Oly. | Putnam-AXIOM |
> |-|-|-|-|-|-|
> |base| 45.0+-2.3 | 68.2+-1.5 | 21.3+-1.2 | 36.2+-1.2 | 6.7+-1.4 |
> | GRPO | 59.2+-1.4 | 80.3+-1.3 | 30.5+-1.3 | 41.1+-1.1 | 15.6+-1.2 |
> | RiskPO | 61.3+-1.2 | 81.4+-2.2 | 32.3+-1.3 | 42.8+-1.2 | 17.2+-1.1 |
>
> > *[Q2]. What does “papers” refer to in lines 251 and 252?*
>
> **RQ2**
> We apologize for not making a clear definition of "paper" in the manuscript. The bundle refers to the group of $B$ different questions. When the model generate $G$ responses per-question, a paper denotes one sampled combination where we pick exactly one response for each of the $B$ questions in the bundle. We will clarify this terminology in the revised version.
>
>
> > *[Q3]. I think the connection between Sec. 5 and Secs. 3,4 is somewhat weak. Could you further explain why strengthening the gradients on hard questions can address the issue of overconfident convergence?*
>
> Thank you for this question. **Sections 3–4 introduce the RiskPO objective and algorithm (MVaR over bundle scores and the corresponding advantage), while Section 5 explains why this particular design mitigates overconfident convergence.** Specifically, Proposition 1 in Section 5 links the per-step entropy change to the covariance between log-probability and the advantage, and Assumption 1 (supported by Fig. 3/9) captures the empirical monotonicity between reward and log-probability. Under this assumption, Theorem 2 shows that our MVaR-based advantage yields a strictly smaller covariance term than the mean-based GRPO advantage, implying slower entropy decay and reduced entropy collapse for RiskPO. We will clarify this logical chain more explicitly in the revised version.
>
> Intuitively, **the MVaR objective in RiskPO not only changes the final optimum but also changes how the policy explores during training,** because RL updates are driven by on-policy rollouts. If the objective has little discriminative power on hard questions or places too much effective weight on easy questions, the policy quickly focuses on easy cases, repeatedly rolling out and reinforcing high-probability but shallow reasoning paths, which leads to overconfident convergence and entropy collapse. By giving hard (low- and medium-reward) bundles a stronger and more persistent influence in the objective, RiskPO forces the policy to keep allocating probability mass to challenging regions, thereby counteracting this collapse and leading to a more calibrated, higher-entropy policy.

---

### Official Review · Reviewer_GymV · 2025-11-03

**Soundness:** 3
**Presentation:** 4
**Contribution:** 3
**Rating:** 8
**Confidence:** 3

**Summary:**

This paper proposes a new RL algorithm with verifiable rewards. The authors argue that GRPO mainly optimize mean rewards, leading to model overfitting to easy prompts and entropy collapse. The authors propose to replace mean-based objectives with risk-sensitive measures that emphasizing the lower-tail of the reward distribution. To obtain distributional information on the reward, authors propose to group multiple prompts and compute a shared reward distribution. With extensive experiments, the authors demonstrate that the proposed method improves performance across reasoning tasks compared to GRPO.

**Strengths:**

* Novel design: This work proposes a novel RL objective that leverages distributional information of reward to effectively enhance model's capability in solving difficult tasks.
* The work is very well written, presenting complex ideas with clarity that makes both the theoretical analysis and experimental results easy to follow.

**Weaknesses:**

* The proposed framework adds complexity to RL training with additional hyper-parameters to tune.

**Questions:**

NA

---

> ### Author Response · Authors · 2025-11-22
> **Rebuttal**
>
> > *[W1]. The proposed framework adds complexity to RL training with additional hyper-parameters to tune.*
>
> **RW1**
> Thank you for raising this concern. In practice, **RiskPO introduces very little additional tuning burden compared to standard RL training.** For the bundle size, the quantile levels, and the MVaR weight $\omega$, we conduct ablation studies to select reasonable hyper-parameter values, as reported in Tables 3, 4, and 5. These results show that RiskPO is quite robust to the choice of these hyper-parameters, and noticeable performance degradation only occurs under extreme settings.
>
> **The only extra hyper-parameters specific to RiskPO are those used for tracking the reward distribution:** the initial learning rate and decay schedule for the quantile (MVaR) estimators. All other training hyper-parameters (optimizer, policy learning rate, clipping coefficients, batch sizes, etc.) are inherited directly from the GRPO baseline. **Moreover, the quantile updates are quite insensitive to the magnitude of their learning rate:** as long as the step size is within a reasonable range, the estimated quantiles converge smoothly and the overall training behavior is unchanged. In our experiments, we use a single default choice of these quantile-related hyper-parameters across all datasets and models, without per-task retuning.
>
> From a stochastic-approximation theory perspective, optimization with quantile tracking follows a standard two-time-scale framework whose convergence and finite-time behavior have been extensively studied. The basic convergence requirements are quite mild, requiring only additional conditions such as $\sum_k \gamma_k = \infty$ and $\sum_k \gamma_k^2 \lt \infty$, and $\eta_k=o(\gamma_k)$, where $\eta_k$ is the learning rate for models' parameters and $\gamma_k$ is for tracking the quantiles. For achieving theoretically optimal convergence rates, existing results on multi-time-scale stochastic approximation [1] further provide explicit step-size forms, such as $\gamma_k = \gamma_0 k^{-2/3}$ and $\eta_k = \eta_0 k^{-1}$. These theoretically grounded scheduleing references greatly reduces the amount of hyperparameter tuning needed in practice.
>
> On the implementation side, RiskPO only requires maintaining a small vector that tracks a few key quantiles of the reward (or bundle return) distribution and comparing these tracked values with the sampled returns to form the MVaR-based advantage. These operations are negligible compared with the cost of LLM forward passes and backpropagation. The quantile tracker can be implemented as a lightweight “reward post-processing” module that sits between the reward model and the policy gradient step, making the method essentially plug-and-play within existing RLVR training pipelines.
>
> [1] Doan, Thinh T. "Nonlinear two-time-scale stochastic approximation: Convergence and finite-time performance." IEEE Transactions on Automatic Control 68.8 (2022): 4695-4705.

---

### Official Review · Reviewer_TQ2d · 2025-11-04

**Soundness:** 2
**Presentation:** 3
**Contribution:** 2
**Rating:** 4
**Confidence:** 3

**Summary:**

The paper proposes a risk-sensitive alternative to mean-based RLVR (Reinforcement Learning with Verifiable Reward) objectives. It introduces a Mixed Value-at-Risk (MVaR) loss that emphasizes the lower tail of the reward distribution and a “bundling” trick that aggregates multiple questions to avoid vanishing advantages when all samples for a question are wrong. The theory links advantage–logprob covariance to entropy changes and argues MVaR mitigates entropy collapse. Empirically, the paper reports Pass@k gains over GRPO and recent variants on several math benchmarks, plus smaller gains on coding and multimodal tasks.

**Strengths:**

Motivation is reasonable: mean objectives over-weight easy, frequent modes; a lower-tail-aware loss could push exploration of harder cases.

Simple to implement: the method is a drop-in change atop GRPO-like training (quantile tracking + bundle-level advantage).

Some positive results: the paper reports consistent improvements on hard math suites and provides ablations (quantiles, bundle size, risk-seeking vs risk-averse).

**Weaknesses:**

Novelty feels incremental.
Applying standard risk measures (CVaR/RVaR/MVaR) to RLVR is a natural extension, and the “bundling” device is an engineering fix to sparse binary rewards. The paper would read more convincingly if it isolated how much of the gain comes from risk weighting versus bundling. A missing baseline is Mean + Bundling (i.e., GRPO-style mean objective but with bundles). A direct CVaR-only baseline adapted to RLVR is also needed to understand if “mixed” VaR really matters.

Benchmark choice is dated and misses stronger generalization tests.
MATH500 and GSM8K are saturated; GSM8K-Platinum exists, and Putnam-AXIOM (arXiv:2508.08292) directly targets extrapolative generalization with functionally novel problems. If the claim is “expands the reasoning frontier,” I would expect evaluation on Putnam-AXIOM and related unseen-variation splits. As written, improvements on older suites are less persuasive.

Catastrophic forgetting is not ruled out.
RL-trained reasoning models are known to regress on distributional pockets they previously handled well. The paper attributes entropy dynamics to the objective, but the same curves could reflect forgetting. Please include a retention analysis: pre/post accuracy on held-out skills or difficulty bins (and on easier sets that should not degrade), plus calibration/entropy before/after. Without this, the claimed mechanism is under-identified.

Prompt-level diversity controls are a missing baseline.
There is emerging evidence that you can preserve output diversity/entropy via prompted diversity instructions (e.g., “enumerate diverse solution paths; assign probabilities”) and light sampling changes—no RL required. Please include a strong prompt-only diversity baseline, matched on compute, to test whether RiskPO’s benefits exceed careful prompting.

Figure 1 is not convincing; statistics are thin elsewhere.
The AIME learning curves (Pass@32/Avg@32) are noisy and, as plotted, do not clearly outperform GRPO in a way that rules out randomness; the panel lacks confidence intervals. AIME’s n=30 also makes variance high. Later figures (e.g., Figure 4) are more favorable but still fairly close on some panels. Please add multi-seed CIs, paired bootstrap on Pass@k, and exact tests (e.g., permutation) for headline numbers across all datasets, and report standardized effect sizes. For small suites (AIME), de-emphasize them in the main paper or aggregate multiple seeds/splits.

Theory–practice gap.
The entropy analysis uses a tabular softmax and a natural-gradient step; actual training is transformer + Adam with sequence-level clipping and IS. While this is common in theory, please add practical diagnostics (token-level entropy, solution-path diversity, coverage metrics) to demonstrate that the claimed mechanism holds in the real training dynamics observed here.

Verifier details and noise sensitivity.
Because RLVR relies on automatic verifiers, please specify the exact checkers per dataset (math normalization rules, code execution policy, multimodal grading) and include a robustness check to verifier noise. Tail-emphasizing objectives can amplify reward mislabels.

**Questions:**

Specific requests / experiments that would change my mind

Add Putnam-AXIOM (and other unseen-variation splits) with 95% CIs and p-values; show RiskPO beats strong GRPO/DAPO/GSPO baselines and a prompt-diversity baseline at matched compute.

Ablate attribution: (a) Mean + Bundling, (b) CVaR-only + Bundling, (c) MVaR without bundling.

Forgetfulness audit: pre/post retention on easy skills and previously-mastered bins; report calibration changes.

Statistics: ≥5 seeds, paired bootstrap CIs for Pass@k and Avg@k, and a preregistered evaluation script.

Compute parity: confirm identical budgets, temperatures, decoding, and training steps for all baselines.

Prompt baseline: add a well-tuned diversity-prompting method that explicitly enumerates multiple reasoning paths with probability assignment, to test whether RL is necessary for the claimed effects.

Minor comments

Clearly define Avg@k in the main text; it is referenced but not formalized.

Explain the length-normalized sequence IS ratio exponent (the 1/|y| in Eq. (3)): why this choice? Any stability issues without it?

Acknowledge AIME’s small-n variance in-paper (not just in the appendix) and report CIs in all main figures.

Tighten text and fix minor wording (“catalyse”, “papers” vs “bundles” in Algorithm 1 Step 6).


Note: all writing was assisted by LLMs, but all the thoughts were my own.

---

> ### Author Response · Authors · 2025-11-22
> **Rebuttal part 1**
>
> > *[W1]. Novelty feels incremental.*
>
> **RW1**
> It does exist some literature that proposes CVaR-based policy gradient algorithms in the classical RL setting. Inspired by these works, we propose the MVaR-based RiskPO and argue that our MVaR criterion is new within the RLVR literature. Prior works mainly focus on improving the mean-based methods, but rather consider to investigate the RLVR from different optimization objective. **We are the first to introduce the risk-sensitive policy optimization to the RLVR.** Moreover, the novelty of RiskPO remains evident even in the broader context of distributional RL. The proposed MVaR-objective policy optimization framework enables flexible attention allocation to different regions of the return distribution. It also provides a unified perspective, since **both the mean objective and the CVaR objective emerge as special cases under MVaR.** Through theoretical and empirical analysis, we highlight the limitations of prior methods and deliver a well-supported risk-averse learning paradigm.
>
> We note that maximizing the mean reward on single questions is only a proxy for the true RLVR goal of improving test-time metrics such as Pass@k. Our MVaR + bundling objective strictly generalizes GRPO-style mean-based methods: when $\alpha=0, \beta=1$, and the bundle size $B=1$, RiskPO reduces to the standard formulation. It subsumes the original objective while enabling richer risk-sensitive behavior and, with suitable hyperparameters, can match or improve upon GRPO.
>
> > *[W3] and [Q3]. Catastrophic forgetting is not ruled out. Forgetfulness audit: pre/post retention on easy skills and previously-mastered bins; report calibration changes.*
>
> **RW3 and RQ3**
> The MATH500 and Olympiad datasets have the most questions. So we partition the two datasets to investigate whether the RiskPO results in catastrophic forgetting. We prompts the base model for 16 times for each question, calculate the success rate, and partition the dataset into two part with the same number of question according to the success rate. We report the pass@1 accuracy and the calibrated output entropy on the different parts of datasets (the entropy value is shown within the parentheses). The part one is the half with higher success rate, and the part two is the half with lower success rate. As shown in the below table, **the GRPO and RiskPO has similar performance on the part one**, but the entropy of RiskPO is higher than GRPO. It suggests that RiskPO do not cause catatrophic forgetting and achieving improvement on easy question do not need to exhaust the entropy. **In the part two, the RiskPO has clear advantage over the GRPO**, since gaining improvement on the hard problems requires sufficient exploration. This experiment proves that the RiskPO achieve consistently improvement on hard questions while avoiding catastrophic forgetting on easy questions.
>
> | method | MATH500 (part one) | MATH500 (part two) |
> |-|-|-|
> | base | 82.3+-2.6 (0.38) | 37.6+-2.5 (0.35) |
> | GRPO | 97.6+-1.2 (0.08) | 60.8+-1.8 (0.16) |
> | RiskPO | 97.8+-1.2 (0.13) | 65.8+-1.4 (0.22) |
>
> | method | Olympiad (part one) | Olympiad (part two) |
> |-|-|-|
> | base | 42.8+-2.5 (0.29) | 18.3+-2.3 (0.32) |
> | GRPO | 54.4+-1.1 (0.04) | 25.3+-1.8 (0.10) |
> | RiskPO | 54.4+-1.2 (0.13) | 28.6+-2.1 (0.18) |

---

> ### Author Response · Authors · 2025-11-22
> **Rebuttal part 2**
>
> > *[W4] and [Q6]. Prompt-level diversity controls are a missing baseline. Prompt baseline: add a well-tuned diversity-prompting method.*
>
> **RW4 and RQ6**
> The original prompt we used for evaluating is:
>
> *"Please reason step by step, and put your final answer within \\boxed{}."*
>
> We design a prompt-level diversity control scheme to encourage the model to reason with variety. The model is use the following prompt:
>
> *"You are an expert math problem solver. First, enumerate 3 diverse solution paths to solve the problem. For each path, clearly state the key idea and give a probability that this path leads to a correct final answer. Then, choose the most reliable path and carry it out in detail. Finally, output the final answer in the \\boxed{} on a separate line."*
>
> Then we evaluate the base model with the designed prompt, and the GRPO trained models with the prompt to investigate how prompt-level diversity controls would affect the performance. As shown in the following table, the base model indeed benefits from the diverisity-encouraged prompt, but the performance is not comparable with RiskPO. We also combine the prompt with the model trained by GRPO. **The improvement of the prompt is incremental compared with the improvement RiskPO brings to the base model.** We argue that the entropy collapse in GRPO hinder the model's exploration to acquire stronger reasoning ability. The prompt-level control is not a fundamental solution, even though it brings some advantages.
>
> | method | AMC | MATH500 | Minerva | Oly. | Putnam-AXIOM |
> |-|-|-|-|-|-|
> | base | 32.6+-1.5 | 59.8+-1.2 | 20.5+-1.2 | 30.4+-1.3 | 2.5+-2.1 |
> | base+prompt engineer | 36.3+-2.2 | 63.7+-1.4 | 22.7+-1.3 | 33.1+-0.3 | 3.9+-0.6 |
> | GRPO | 56.5+-1.3 | 79.4+-1.4 | 27.0+-1.2 | 39.6+-1.2 | 14.1+-0.8 |
> | GRPO+prompt engineer | 57.1+-1.2 | 79.6+-1.3 | 27.8+-1.2 | 40.2+-2.1 | 15.0+-1.4 |
> | RiskPO | 60.9+-1.0 | 81.7+-1.3 | 29.7+-1.3 | 41.3+-1.2 | 17.8+-1.5 |
>
> > *[W2], [W5], [Q1], and [Q4]. Benchmark choice is dated and misses stronger generalization tests. statistics are thin elsewhere. Concerns about Statistics. Benchmark choice is dated and misses stronger generalization tests. Add Putnam-AXIOM (and other unseen-variation splits) with 95% CIs and p-values.*
>
> **RW2, RW5, Q1, and RQ4**
> Due to the limited computation resources, we can not afford to train the model on the hard-level math task for multiple times. We train the model once and use the model to generate multiple times in the evaluation phase to cancel out the randomness. In the original paper, we report the Pass@k metric from evaluating the models with three different seeds (23,66,87) and taking the average. **To ensure the statistics significance, we evaluate each model five times with different seeds (23,66,87,26,15) and report 95% confidence intervals in the +- half-width format.** We add the Putnam-AXIOM in our evaluation results in the following table. We evaluate our models on the Functional Variations (265 problems) of the Putnam-AXIOM. It appears that Putnam-AXIOM is the hardest dataset, and our RiskPO still achieve the best performance on it.
> | method | AIME25 | AIME24 | AMC | MATH500 | Minerva | Oly. | Putnam-AXIOM |
> |-|-|-|-|-|-|-|-|
> | GRPO | 21.8+-2.2 | 21.3+-2.2 | 56.5+-1.3 | 79.4+-1.4 | 27.0+-1.2 | 39.6+-1.2 | 14.1+-0.8 |
> | DAPO | 29.3+-1.9 | 29.3+-1.9 | 58.6+-0.8 | 78.5+-1.3 | 29.2+-1.3 | 40.6+-0.9 | 15.3+-1.3 |
> | GSPO | 30.0+-2.9 | 30.7+-1.8 | 59.2+-0.6 | 80.1+-1.5 | 28.8+-1.4 | 40.2+-1.2 | 16.5+-1.6 |
> | RiskPO | 34.0+-3.5 | 34.0+-3.5 | 60.9+-1.0 | 81.7+-1.3 | 29.7+-1.3 | 41.3+-1.2 | 17.8+-1.5 |
>
> We **have updated all the Pass@k and Avg@k figures** in the paper with confidence intervals according to our additional evaluation results.
>
> > *[W6]. Theory–practice gap.*
>
> **RW6**
> Thanks for raising the concern about the theory and practice gap. Even though our analysis is conducted in the simplified setting, the theoretical results can be verified by empirical evidence. To bridge the theory–practice gap, **we added new diagnostics in Appendix A.4 computed under the actual transformer + Adam training setup on the easy-level math task**. For both GRPO and RiskPO, we explicitly track **(i) the covariance between log-probabilities and advantages and (ii) the one-step change of average token-level entropy.** As shown in Fig. 9, these two quantities move in lockstep, empirically validating Proposition 1, and the covariance of RiskPO is consistently smaller than that of GRPO throughout training, in line with Theorem 2. Together with the Pass@k / Avg@k curves (Figs. 4, 7–8), which reflect improved coverage and solution-path diversity at larger k, these diagnostics demonstrate that the entropy mechanism we analyze theoretically also holds in the real training dynamics of our transformer-based implementation.

---

> ### Author Response · Authors · 2025-11-22
> **Rebuttal part 3**
>
> > *[W7]. Verifier details and noise sensitivity.*
>
> **RW7**
> For the math task, we adopt an RLVR-style rule-based math verifier that first aggressively normalizes both predictions and references (stripping LaTeX wrappers, unifying common macros and symbols such as all fractions to \frac and π→pi, removing units/currency, standardizing numbers, intervals, and percentage conventions). On top of this, it applies a layered equality check: lenient string matching, then numeric comparison via math.isclose with a relative tolerance of 1e-4 (also considering {ref/100, ref, ref*100} for percentages), and finally symbolic comparison using SymPy by checking whether simplify(a - b) == 0 or their numerical evaluations coincide within the same tolerance under timeouts. The verifier also supports structured answers such as intervals, tuples, and matrices via element-wise checking with the same logic, and enforces reduced-form fractions (e.g., accepting 1/2 but not 2/4 as equivalent).
>
> For the code task, we enforces a unified format with <think>... followed by the final python ... block and extracts only the last fenced code block; if no valid block is found, the attempt is marked as a format error. The extracted code is executed in an isolated subprocess with ~1GB memory limit, per-call timeouts, a temporary working directory, cleaned environment variables. For correctness, we support both functional tests (running user code plus test code with asserts, with standard imports and LeetCode-style helper classes) and input/output tests (sampling up to 16 cases during training, using all cases at evaluation, running in parallel threads); LiveCodeBench tasks are handled in a separate process with a global timeout. Outputs are compared with several increasingly tolerant rules (direct equality, list/tuple normalization, np.allclose for floats, and set-based comparison for unordered text), which together reduce format-induced verifier noise while keeping the execution policy conservative and safe.
>
> For multimodal reasoning (e.g., Geo3K) we use essentially the same verifier pipeline as for math. Concretely, we rely on mathruler’s grade_answer() (and extract_boxed_content() when use_boxed=True) to compute an accuracy reward acc_reward by comparing the model answer against the ground truth; optionally we also compute a format reward format_reward based on whether the output contains the required <think>...</think> and \boxed{...} structure. In our experiments, we require the verifier ignores formatting and returns a binary 0/1 score purely based on answer correctness, but the underlying grading logic is the same as for the math verifier.

---

> ### Author Response · Authors · 2025-11-22
> **Rebuttal part 4**
>
> > *[Q2]. Ablate attribution: (a) Mean + Bundling, (b) CVaR-only + Bundling, (c ) MVaR without bundling.*
>
> **RQ2**
> We provides additional ablations about differents objective and bundling. The bundling technique is specifically tailored for our risk-sensitive objective to enrich the distributional information. For mean-based objective, the bundling technique should not bring significant advantage. As shown in the below table, the MVaR with bundling performs the best and the CVaR with bundling is the second, since a vanilla CVaR do not place extra attention on the lower tail of reward distribution even though it avoid over-emphasizing the upper tail which would cause entropy collapse. **Comparing the GRPO with its bundling version, it shows that the bundling technique do not bring benefits to the mean-based method,** which coincide with our intuition. Our explanation is that the mean only reflects partial information of a distribution, and enrich the distribution can not bring advantages to mean-based method. The MVaR without bundling performs the worst, which suggests that the bundling is necessary for distribution-level algorithm.
>
> | method | GSM8K-Platinum | MATH |
> |-|-|-|
> | Mean(vanilla GRPO) | 78.9+-1.2 | 54.5+-1.2 |
> | Mean + Bundling | 79.0+-1.1 | 54.6+-1.3 |
> | CVaR-only + Bundling | 80.7+-0.9 | 56.0+-1.1 |
> | MVaR without bundling | 78.6+-1.2 | 53.8+-1.3 |
> | MVaR with bundling(RiskPO) | 82.6+-1.2 | 56.8+-0.8 |
>
> > *[Q5]. Compute parity: confirm identical budgets, temperatures, decoding, and training steps for all baselines.*
>
> **RQ5**
>
> We thank the reviewer for raising this point. In our experiments, all methods that we train within our RLVR framework (GRPO, RiskPO, and the risk-seeking variant) are run under the same compute budget and decoding configuration. Concretely, for each task we use the same base model, the same training datasets (DAPO-MATH-17K for hard math, MATH+GSM8K for easy math, Geo3K for multimodal, and Archer-6K for code), and the same RL training schedule: identical numbers of updates (500 steps for hard math, 200 steps for easy math, 300 steps for code, and 100 steps for multimodal), batch size (512), mini-batch size (128), and sampling budget per update (we generate 10 trajectories per problem). We also share the same optimization hyperparameters across these methods: clipping threshold $\epsilon=0.2$ (for DAPO we set $\epsilon_{low}=0.2, \epsilon_{high}=0.28$), no KL penalty and no entropy regularization, and the same FSDP-based training backend.
>
> At both training and evaluation time, all models are decoded with the same sampling configuration: we use vLLM as the inference backend, temperature 0.8, top-p=1.0, and a maximum output length of 3072 tokens. These settings are applied uniformly to all baselines and RiskPO when computing Pass@k and Avg@k on all benchmarks.
>
> > *[MC]. Explain the length-normalized sequence IS ratio exponent*
>
> **RMC**
> We draw inspiration from GSPO, and apply a length-normalized sequence IS ratio[1]. The normalization makes the ratio essentially length-invariant: short and long solutions are re-scaled to a comparable per-token scale, so that a fixed clipping range does not implicitly favour one length regime over another. For long math solutions this leads to extremely heavy-tailed importance weights and high-variance gradients unless one applies very aggressive clipping. Using the length-normalized exponent is a standard “tempering’’ of IS ratios: it preserves the ordering of samples (the transformation is monotone), but keeps the magnitude of the ratios under control and avoids training instabilities that would otherwise arise from a few very long trajectories dominating the update.
>
> [1] Zheng, Chujie, et al. "Group sequence policy optimization." arXiv preprint arXiv:2507.18071 (2025).

---

### Author Response · Authors · 2025-11-30
**General Responses**

We thank the AC and all four reviewers for their careful and constructive feedback. Our paper proposes RiskPO, a risk-sensitive policy-gradient framework that optimizes a Mixed-Value-at-Risk (MVaR) objective over bundle-wise returns for RLVR; conceptually, it views RLVR as optimizing a general distributional risk functional with GRPO as the special mean-based case, thereby preserving GRPO’s strengths while improving reasoning performance and mitigating entropy collapse.

**Novelty and motivation.** Several reviewers (TQ2d and Hp4w) questioned whether the contribution is incremental relative to CVaR-based policy gradients or GRPO. In the rebuttal we make clarification. GRPO and related RLVR methods are mean-based in the sense that their underlying risk functional is the expected reward; the use of (group-wise) advantages is purely a variance-reduction device and does not alter this objective in expectation. RiskPO instead **optimizes a genuinely non-linear distributional functional** over bundle returns (MVaR), which downweights the upper tail, emphasizes the lower tail, and recovers both mean and CVaR as special cases. Moreover, **the proposed bundling mechanism is fundamentally different from GRPO’s grouping:** GRPO groups multiple responses to the same question to construct a baseline, whereas RiskPO forms bundles over different questions and defines advantages on the distribution of bundle scores. This moves RLVR from per-question 0/1 rewards to a richer distribution over multi-level returns and is crucial for our risk-sensitive design.

**Theory and its connection to practice.** A key concern from Reviewer TQ2d and Hp4w was whether our theoretical analysis (Section 5) truly explains the empirical behavior of GRPO and RiskPO. We restate our theoretical part and add supplementary experiment to further justify our theory. First, we clarify the Proposition 1: the entropy change is governed by the covariance between log-probabilities and advantages. Based on a mild monotonicity assumption (empirically validated in Figs. 3 and 9), we show that the MVaR-based advantages systematically reduce this covariance (Theorem 2), since they explicitly downweight the upper tail and focus on low/medium reward regions, thereby mitigating entropy collapse. Finally, we **add new diagnostics under the actual transformer + Adam setup in the the revision of the paper**: we track (i) covariance between log-probability and advantage, and (ii) per-step entropy change, and show these move in lockstep as predicted, with RiskPO consistently exhibiting smaller covariance and slower entropy decay than GRPO.

**Expanded experiments, statistics, algorithms details.** Reviewers (TQ2d, yqFg and Hp4w) raised concerns about benchmark coverage, statistical rigor, and additional ablations. We makes explicit responses and add new experiments. We **re-evaluate all methods with five seeds and now report 95% confidence intervals** (+- half-width) for Pass@k / Avg@k across all benchmarks (figures with CI regions are in the revision). We expanded our evaluation to both the hardest-level Putnam-AXIOM benchmark and a larger DeepSeek-R1-Distill-Qwen-7B model, where RiskPO consistently outperforms GRPO and other strong baselines across math datasets. We **implemented a strong prompt-engineering-based diversity control** and evaluated it on the base model and GRPO. While this prompt yields modest gains, its improvements are clearly subsumed by RiskPO, indicating that **RiskPO addresses the entropy issues that cannot be solved purely at the prompting level.** We perform extensive ablations over quantile/risk levels, as well as mean + bundling, CVaR-only + bundling, MVaR without bundling, and MVaR with bundling. These show that our full design (MVaR + bundling) is consistently strongest and that bundling is indeed necessary for leveraging distributional information in risk-sensitive objectives.

**Other concerns.** To address concerns about catastrophic forgetting (Reviewer TQ2d), we **partition MATH500 and Olympiad into easy and hard halves based on the base model’s success rate, and report Pass@1 + entropy on each half.** The results show that on easy questions, GRPO and RiskPO achieve similar accuracy, but RiskPO maintains substantially higher entropy, indicating it avoids overconfident collapse. On hard questions, RiskPO has clear accuracy gains over GRPO and exhibits less entropy collapse, confirming that it truly improves performance on difficult problems instead of overfitting easy ones. The Reviewer GymV questioned the added complexity and tuning burden of RiskPO. We clarify that **RiskPO only adds a lightweight quantile-tracking module on top of GRPO,** incurring negligible extra computation and almost no additional tuning overhead.

We hope these clarifications help the AC see that RiskPO offers a novel, empirically validated, and practically usable risk-sensitive alternative to existing mean-based RLVR methods.

---

### Meta-Review · Area_Chair_BFQ8 · 2026-01-08

**Summary:**

This paper proposes RiskPO, a risk-sensitive reinforcement learning framework for LLM post-training that replaces mean-based objectives (e.g., GRPO) with a Mixed Value-at-Risk (MVaR) objective. The method includes a bundling scheme that aggregates multiple questions to enrich the binary reward signal. The paper provides theoretical analysis linking risk-averse updates to entropy preservation and demonstrates empirical improvements on mathematical reasoning, code generation, and multi-modal benchmarks.

This paper studies an interesting idea and presents comprehensive experiments to support it. The major outstanding concern from reviewers about this paper is novelty. While reading this paper, I noticed a discrepancy between the reported base model (DeepSeek-R1-Distill-Qwen-1.5B) performance on AIME 2024 and official numbers (https://huggingface.co/deepseek-ai/DeepSeek-R1-Distill-Qwen-1.5B), which may warrant clarification regarding the evaluation protocol.

Overall, I think this paper is a borderline accept for ICLR, as the comprehensive experiments and theoretical analysis provide sufficient contribution despite the novelty concerns.

**Reviewer Concerns:**

I think most experimental concerns, including experimental details and required additional experiments are addressed. Outstanding concerns are mainly about novelty.

**Reviewer Scores:**

Reviewer TQ2d = 4; Reviewer GymV = 8; Reviewer Hp4w 50% = 4, 50% 4 -> 6; Reviewer yqFg = 6

---

### Decision · Program_Chairs · 2026-01-26

Accept (Poster)